# Primary Composition of Kimberlite Melt

Sergey Kostrovitsky [1,2,*], Anna Dymshits [2], Dmitry Yakovlev [1], Jing Sun [3], Tatiana Kalashnikova [1,2], Igor Ashchepkov [4] and Olga Belozerova [1]

1   Institute of Geochemistry SB RAS, Irkutsk 664033, Russia; dimasiky0@yandex.ru (D.Y.); kalashnikova@igc.irk.ru (T.K.)
2   Institute of Earth's Crust SB RAS, Irkutsk 664033, Russia; adymshits@crust.irk.ru
3   State Key Laboratory of Petroleum Resources and Engineering, Beijing 102249, China; sunjingvv@163.com
4   Institute of Geology and Mineralogy SB RAS, Novosibirsk 630090, Russia; igor.ashchepkov@igm.nsc.ru
*   Correspondence: serkost@igc.irk.ru

**Abstract:** The compositions (mineralogy, major- and trace-element chemistry of rocks and minerals, and Sr-Nd-Hf isotope systematics) of two kimberlite bodies, the Obnazhennaya pipe and the Velikan dyke from the Kuoika field, Yakutian kimberlite province (YaKP), which are close to each other (1 km distance) and of the same Upper Jurassic age, are presented. The kimberlites of the two bodies are contrastingly different in composition. The Obnazhennaya pipe is composed of pyroclastic kimberlite of high Mg and low Ti composition and is characterized by high saturation of clastic material of the lithospheric mantle (CMLM). The pyroclastic kimberlite contains rare inclusions of coherent kimberlite from previous intrusion phases. The Velikan dyke is represented by coherent kimberlite of relatively high Fe and high Ti composition, having neither mantle xenoliths nor olivine xenocrysts. The similarity of the isotopic geochemical characteristics for kimberlites from both bodies and their spatial and temporal proximity suggest that their formation is associated with the presence of a single primary magmatic source located in the asthenosphere. It is proposed that the asthenospheric melt differentiated into two parts: (1) a predominantly carbonate composition and (2) a carbonate–silicate composition, which, respectively, formed (a) low Fe and (b) Mg-Fe and high Fe-Ti petrochemical types of kimberlites. Both parts of the melt had different capabilities to capture the xenogenic material of the mantle rocks. The greater ability to destroy and, subsequently, capture CMLM belongs to the melt, which formed a high Mg type of kimberlite and which, according to the structural–textural classification, more often corresponds to the pyroclastic kimberlite. It is suggested that the primary kimberlite melt of asthenospheric origin is similar in composition to the high Fe, high Ti, coherent kimberlite from the Velikan dyke (in wt. %: $SiO_2$–21.8, $TiO_2$–3.5, $Al_2O_3$–4.0, FeO–10.6, MnO–0.19, MgO–21.0, CaO–17.2, $Na_2O$–0.24, $K_2O$–0.78, $P_2O_5$–0.99, $CO_2$–12.6). It is concluded that the pyroclastic kimberlite contains only xenogenic Ol, whereas some of the Ol macrocrysts with high FeO content in the coherent kimberlite have crystallized from the melt. The similarity of Sr-Nd-Hf isotope systematics and trace element compositions for kimberlites of different ages (from Devonian to Upper Jurassic) in different parts of the YaKP (in the Kuoika, Daldyn and Mirny fields) indicates a single long-lived homogeneous magmatic asthenospheric source.

**Keywords:** kimberlite; petrochemical type; origin of kimberlites; the disintegrating ability of melt; mineral chemistry; olivine; Sr-Nd-Hf systematics





## 1. Introduction

Understanding the primary composition of kimberlite melt is crucial. It is essential to comprehend the origin of kimberlites and to determine the composition of the asthenospheric melt that our planet produces. The megacryst association begins to crystallize synchronously with the ascent of kimberlite magma, where the asthenospheric melt serves as one of the assumed sources [1–5]. A large number of researchers have devoted their works to kimberlites of the Yakutian kimberlite province (YaKP) [6–10].

Despite significant progress in understanding the nature of kimberlite rocks in recent years [11–21], a number of issues about genesis remain debatable. For example: (1) What was the primary composition of the kimberlite melt [13,19,22,23]? (2) What are the relationships between volcanoclastic (pyroclastic) and coherent kimberlites [24,25]? (3) Is olivine only a xenogeneic mineral [26–28]?

Determining the main composition of kimberlites is challenging due to their hybrid nature and inclusion of clastic material of the lithospheric mantle (CMLM), which has been both captured and partially assimilated. This complexity sets kimberlites apart from most other igneous rocks. Therefore, scientists trying to identify the primary composition of kimberlite melt either eliminate CMLM or select aphyric varieties of kimberlites as a reference sample for study. However, as the kimberlite magma has already assimilated the CMLM, the resultant composition data remain uncertain. Many attempts to determine kimberlites' primary composition have neglected to consider the variety of compositions exhibited in the formation of even one province. Consequently, Roger Mitchell expressed skepticism regarding the basic feasibility of discerning the primary composition of kimberlites [12].

Volcaniclastic (pyroclastic) kimberlite is believed to have formed due to near-surface processes occurring in the conduits of the pipe, such as fluidization or explosions caused by the melt meeting surface water [22,29–32].

Two kimberlite bodies—the Obnazhennaya pipe and the Velikan dyke—were discovered during the study of the northern YaKP field (Kuoika). In this study, the composition of these bodies, along with their spatial and temporal relationships, were used to evaluate the primary composition of the kimberlite melt and to resolve a number of debated issues associated with kimberlite formation. Various scenarios regarding the origin of pyroclastic kimberlite from the Obnazhennaya pipe are discussed in the current study. In addition to pyroclastic kimberlite formed near the surface, the pipe also contains pyroclastic kimberlite that is thought to have originated in the mantle depths.

This article describes the petrography, mineralogy, chemical and isotope geochemical composition of kimberlites of the Obnazhennaya pipe and the Velikan dyke. It should be noted, however, that these bodies are typical in terms of distribution in the Kuoika field, which details the composition of a representative kimberlite composed of high Mg and Fe-Ti petrochemical types.

## 2. General Information about the Obnazhennaya Pipe and Velikan Dyke, and a Petrographic Description

Traditionally, within the Yakutian kimberlite province (YaKP) (Figure 1), the southern diamondiferous fields (including all known primary diamond deposits in Yakutia) and the northern fields (whose kimberlites are mainly characterized by very low diamond content) are distinguished [1–4]. If the pipes and dykes present in the southern fields are kimberlites with low Fe and Fe-Mg petrochemical types [5], then those in the northern fields consist mainly of Fe-Ti-type kimberlites. The peculiarity of the Kuoika field compared to other northern fields is that it contains kimberlites, which include those of low Fe, low Ti and high Fe, high Ti composition [3,6]. The Obnazhennaya pipe and the Velikan dyke, which form two unique rock outcrops, are composed of kimberlites of contrasting compositions.

The Obnazhennaya pipe consists mostly of pyroclastic kimberlite with high Mg, low Ti, whereas the Velikan dyke is composed of coherent kimberlite with high Fe and high Ti. The spatial proximity of these bodies (the distance between them is about 1 km) and the relatively high freshness of the kimberlites (partial preservation of fresh olivine, high content of unaltered mantle xenoliths in the Obnazhennaya pipe [7–10]) make these kimberlite bodies one of the most convenient objects for solving controversial questions of primary kimberlite composition and evolution of the kimberlite melts.

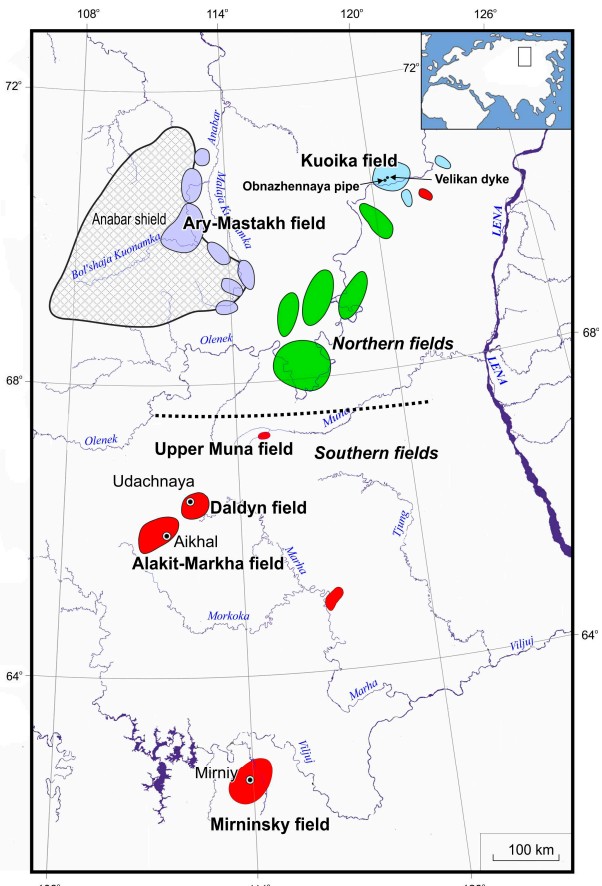

**Figure 1.** Map of Yakutian kimberlite province [4]. The color of the fields corresponds to their age (in million years): green, 428–408; red, 376–344; purple, 229–214; blue, 175–146.

The kimberlite rocks in the Yakutian kimberlite province (YaKP) are typically altered by secondary serpentinization processes, and the pipes are covered by alluvial–deluvial deposits (Mir, Internatsional'naya, Nyurbinskaya, etc.). However, there are rare exceptions in the Kuoika field, particularly the Obnazhennaya pipe and Velikan dyke (Figure 1; Supplementary Figure S1A,B).

The Obnazhennaya pipe has an outcrop up to 20 m high and about 25 m long exposed along the coast of the Kuoika river. It is oval in shape and 30 by 25 m in size. The coordinates of the pipe are N70031′08″, W120030′06″. The sub-vertical Velikan dyke forms a rocky outcrop 3 m thick and 8–10 m high. The strike azimuth of the dyke is 300 NE. The host rocks of the kimberlite bodies are Early Cambrian limestone, dolomite and marl. Estimations suggest that the depth of erosion cutout within the Kuoika kimberlite field varies from 500 [33] to 1900 m [34]. Since the previous assessment of the erosion capacity of the host rocks is founded on an incorrect determination of the lower Palaeozoic age of the pipes in the Kuoika field, it is evidently exaggerated. The erosive section of the Obnazhennaya pipe is likely no more than 1200 m, based on the average of the available estimates. The Upper Jurassic age of the Obnazhennaya pipe was first determined through the discovery of fossils in belemnite rostra [35] and wood [36] in the kimberlites. Using the U-Pb perovskite and rutile method, the Obnazhennaya pipe's age was determined to be 151–154 Ma [37]. The U-Pb method, using perovskite and zircon [37–40], determined that the ages of all 20 kimberlite bodies studied in the Kuoika field are between 145 and 161 Ma. These ages confirm that the Velikan dyke is also from the Upper Jurassic period.

The Obnazhennaya pipe is made mainly of volcanoclastic and pyroclastic kimberlite (as shown in the microphotographs of the thin sections; Figure 2A–F and Supplement Figures S2, S2-1 and S2-2). Volcanoclastic kimberlite is a coarse-grained breccia that fills the

peripheral part of the pipe with a width of 5–10 m, comprising up to 40%–50% debris from country rock. Based on the texture and high content of olivine macrocrysts (>30%), the pyroclastic kimberlite in the Obnazhennaya pipe can be classified as the macrocrystic variety [22,41–43]. The non-genetic term "macrocrystal" indicates the presence of large crystals (>1 mm) of predominantly anhedral olivine and clastic or rounded phlogopite laths. These macrocrysts form via the disintegration of mantle xenoliths (e.g., [22,43–45]). Pyroclastic kimberlite, which fills the central part of the pipe, has a dark green to black appearance within the mesostasis of the massive structure, including 10%–25% country rock fragments. Kimberlite displays a broad array of olivine macrocryst content, ranging from 30% to 50% of the rock volume, while containing a minor quantity of phlogopite (1%–3%). Unaltered Ol macrocrysts (partially (Figure 2A) or completely replaced by serpentine (Figure 2B)) are sub-rounded and angular. There are also individual grains of macrocrysts presented by phlogopite and orthorhombic and monoclinic pyroxene. Sometimes, pyroclastic kimberlite contains very small debris (1–3 mm) of coherent kimberlite featuring microlithic phlogopite or calcite (Figure 2B; Supplement Figures S2-1 and S2-2). The groundmass of the Cal–Srp pyroclastic kimberlite composition consists of cryptocrystalline serpentine, phlogopite, opaque minerals (Ti-Mag) and Ap microcrystals (<1%–2%). Pyroclastic kimberlite contains autoliths of coherent kimberlite up to 10 cm in size, along with xenoliths of rock from the crystalline basement and lithospheric mantle up to 20 cm in size. The majority of mantle xenoliths were discovered within a central, column-like section measuring 3 m × 2 m in size within the outcrop. The heavy fraction of pyroclastic kimberlite minerals from the Obnazhennaya pipe is represented mainly by Ol (85%–95%), Grt (5%–7%), Sp (3%–5%), Cpx (2%–4%) and Opx (1%–2%). In the fine fraction (<0.2 mm), along with the listed minerals, there are individual grains of euhedral crystals of Zrc and Ap.

Three types of coherent kimberlite differing in composition and structure were found in the Obnazhennaya pipe. The most common type of kimberlite is fine- and medium-macrocrystal. It is composed of euhedral or subhedral Ol micro- and macrocrysts in a Srp-Phl (Figure 2C; Supplement Figure S2-3) or, more rarely, Phl-Srp (Figure 2D) groundmass with Mag and Prv unevenly dispersed in amounts ranging from 5% to 15%. Micro- and macrocrysts of unaltered Ol, sometimes slightly corroded in the marginal part, usually contain microcracks over which microcrystalline Phl has developed. The groundmass carbonate is represented by micro- and fine-grained Cal, often microlitic in shape and 0.1 × 0.01 mm in size, which results in a fluid microtexture of the rock [45].

The less common form of kimberlite is autoliths of coherent kimberlite with fine prismatic crystals phlogopite together with calcite in the groundmass (Figure 2E; Supplementary Figure S2-3). Very rare occurrences of type 3 of coherent kimberlite (as illustrated in Figure 2F) consist of 30–70 vol.% obviously clastic and deformed phlogopite macrocrysts in a carbonate groundmass with abundant opaque minerals (Ti–magnetite and perovskite). Additionally, they contain 5–7 vol.% angular and sub-rounded olivine macrocrystals (Figure 2F). Most phlogopite macrocrysts are characterized by signs of deformation and form partly fragmented grains. Such grains are the disintegrated fragments of previously larger phlogopites. Perhaps these autoliths should be attributed to the deep xenoliths of the glimmeritic type.

The Velikan dyke is composed of coherent, finely porphyritic (rarely, aphanitic) dark gray to black kimberlite of a massive texture. The groundmass structure of the of the kimberlite in the Velikan dyke is allotriomorphic granular and micropoikilitic. Olivine microcrysts of 0.1–1 are present within the kimberlite, comprising 3%–15% of the rock's composition. The microcrysts are partially or completely replaced by serpentine. The shape of the microcrysts is oval, idiomorphic and subidiomorphic.

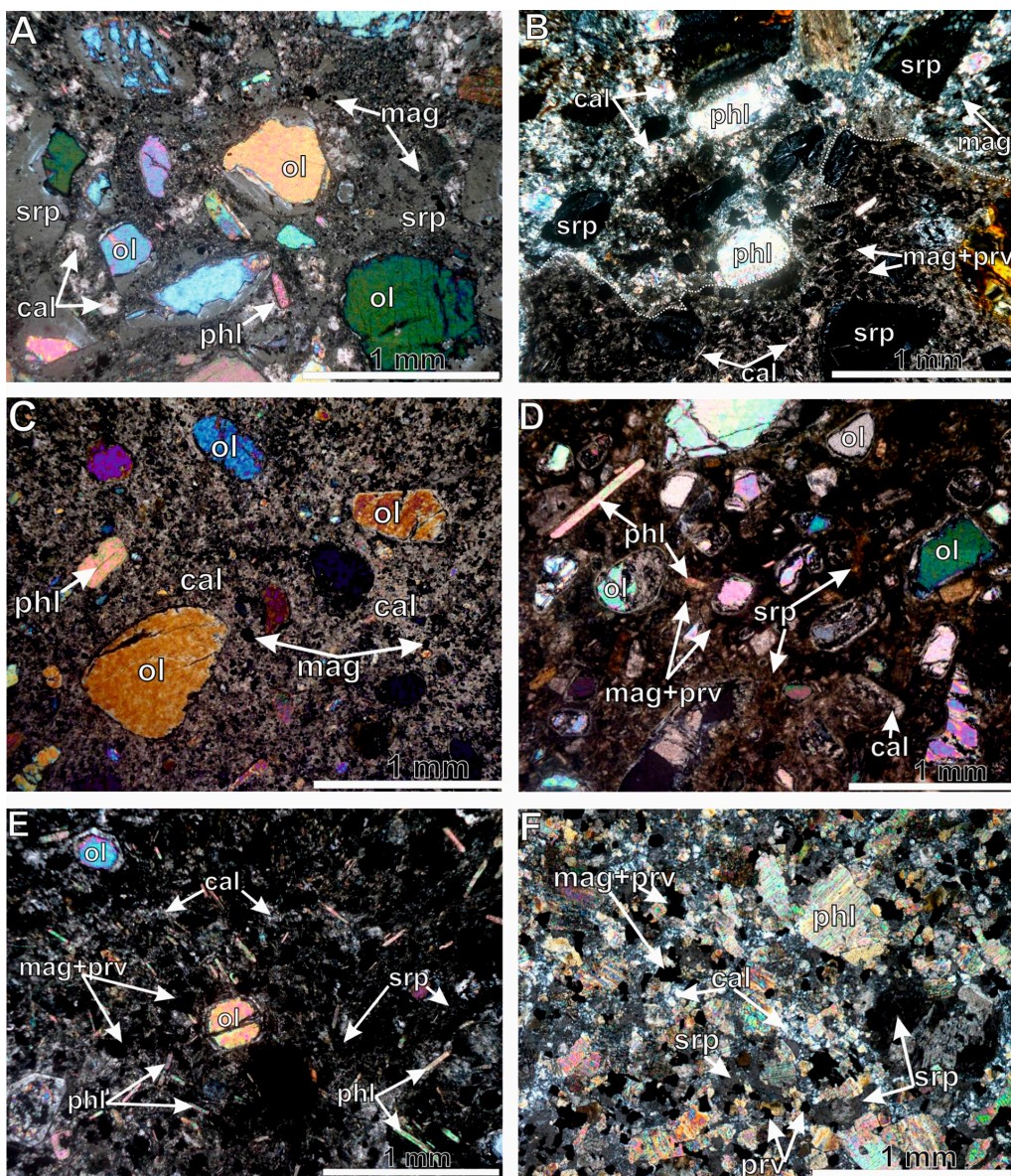

**Figure 2.** The photomicrographs of thin sections of the samples: (**A**)—7–392: pyroclastic kimberlite with essentially serpentine groundmass. Sub-rounded olivine porphyroclasts partly substituted by serpentine, with some preserved relict of fresh olivine. (**B**)—7–269: pyroclastic kimberlite with inclusion of coherent kimberlite containing calcite microcrystals (below). (**C**)—7–384: coherent kimberlite with groundmass of essentially carbonate composition. (**D**)—7–387: coherent kimberlite with groundmass with calcite-serpentine composition. Texture with oriented arrangement of phenocrysts. (**E**)—7–386: fine-grain coherent kimberlite with phlogopite microlites in groundmass. (**F**)—7–388: coherent kimberlite enriched by phlogopite macrocrysts. Abbreviations: olivine—Ol, serpentine—Srp, spinel—Spl, phlogopite—Phl, magnetite—Mag, calcite—Cal.

The groundmass is mainly composed of serpentine–carbonate, including abundant (up to 15%–25%) ore minerals, such as perovskite and Ti–magnetite. A very interesting feature of the groundmass of kimberlite from the Velikan dyke is the atypical arrangement of elongated prismatic apatite crystals, which vary in dimension from $2 \times 10$ to $10 \times 100$ μm and form distinct clusters consisting of over 100 crystals (>100 crystals). The olivine microcrystals sometimes form a fluid texture (Figure 3A). The microcrystals of phlogopite from the groundmass of coherent kimberlite form irregularly distributed fine, thin-lamellar

prismatic crystals (from 50 × 5 to 250 × 80 μm) (Figure 3B,C). A distinctive feature of kimberlite is the absence of olivine macrocrysts.

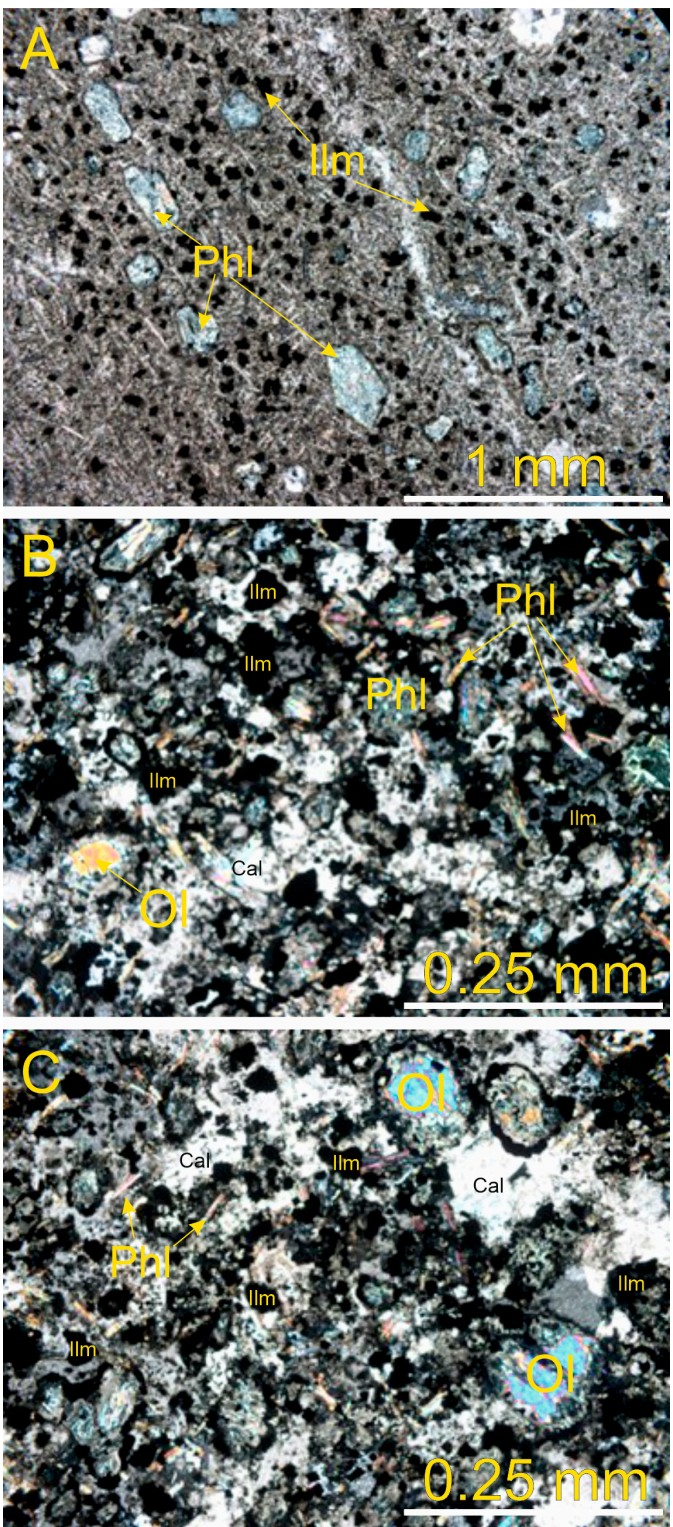

**Figure 3.** Images of the thin sections of the samples from Velikan dyke. (**A**)—the coherent kimberlite with the olivine and phlogopite microcrysts with a fluid texture; (**B**)—the coherent kimberlite; (**C**)—the coherent kimberlite with irregularly distributed fine, thin-lamellar prismatic phlogopite crystals. Abbreviations: olivine—Ol, phlogopite—Phl, ilmenite—Ilm, calcite—Cal.

## 3. Analytical Methods

To describe the petrographic varieties of rocks, we use simplified nomenclature that differentiates only two contrasting types [14,24,25,43,46]: pyroclastic (volcaniclastic) and coherent (hypabyssal) kimberlites with breccia (fragmental) and massive (non-fragmental) textures, respectively. To describe the wide range of chemical composition found within kimberlite rocks, we follow the petrochemical classification system for kimberlite rocks developed by Russian geologists [5,47]. The authors have identified three main petrochemical types of kimberlites: (1) low iron (low-Fe) ($FeO_{total} < 6$ wt. %, $TiO_2 < 1.0$ wt. %); (2) magnesian–ferrous (Mg-Fe) (6–9 wt. % $FeO_{total}$, 1–2.5 wt. % $TiO_2$); (3) ferrous–titanium (Fe-Ti) (8–15 wt. % $FeO_{total}$, 1.5–7 wt. % $TiO_2$). The basis for the development of this classification was the different mineral compositions of the identified types of kimberlites and the different levels of their diamond grade [5].

In this study, we used 30 samples from the Obnazhennaya pipe and the Velikan dyke. The petrographies of the samples were studied in thin sections. The preparation of bulk kimberlite samples, mineral picking and the majority of the analytical work were conducted at the A.P. Vinogradov Institute of Geochemistry (Irkutsk). The whole-rock major-oxides contents were measured by X-ray fluorescence [48] spectroscopy (accuracy = 10 ppm) by an SRM-25 analyzer with a Rh-anode end-window X-ray tube, at an acceleration voltage of 30 kV and a beam current of 40 mA. Calibration was performed against the certified standard samples SGD-1A (gabbro), SI-2 (carbonatite), JB-1 (basalt) and JP-1 (peridotite, Japan).

Trace element abundances in the 30 bulk samples were determined by AAS, flame photometry and XRF methods. In addition, the ICP-MS method was applied to study the full range of trace elements in six samples using the Elan 6100 DRS spectrometer at the Institute of Geochemistry in Irkutsk (analyst: N. Pakhomova). The results of multiple replica measurements of the standards were reproduced with errors (i.e., deviation from accepted standards) less than 5% for Sr, La, Ce, Nd, Sm, Tb, Ho, Er and Yb, and 5%–15% for Y, Zr, Pr, Eu, Gd, Dy, Tm, Lu and Hf, with detection limits of 0.01–0.03 ppm. Meanwhile, Na, K, Li, Rb and Cs were determined by flame photometry (detection limit 1 ppm). The remaining trace elements (Ni, Cr, Cr, V, Zn and Cu) were determined by atomic adsorption spectrometry (AAS), with a detection limit of 10 ppm.

EMPA measurements of the minerals were performed on a JEOL Superprobe JXA-8200 analyzer (Japan) with five wavelength dispersion spectrometers at an accelerating voltage of 20 kV, a beam current of 20 nA, and counting times of 10 s (peak) and 5 s (background) on both sides of the line at the Institute of Geochemistry (analyst: L. Suvorova). The morphology, zoning and relative percentages of the minerals were studied in BSE images of different magnifications obtained on a scanning electron microscope.

The groundmass minerals from the Velikan dyke were studied using a scanning electron microscope (TESCAN, Brno, Czech Republic). The analysis was carried out using an AztecLive Advanced Ultim Max 40 microanalysis system with a nitrogen-free energy dispersive spectrometer (EDS) at an accelerating voltage of 20 kV, a beam intensity of 18.50, an absorbed current of 4.1 nA and a beam diameter of 100 nm.

The Sr, Nd and Hf isotope composition of the kimberlites were analyzed at the Institute of Geology and Geophysics, Chinese Academy of Sciences. The details have been reported in [49,50].

## 4. Results

### 4.1. Major and Rare Elements Chemistry

The major-element composition of the Obnazhennaya kimberlite is represented by 30 whole-rock analyses, including 19 analyses for pyroclastic kimberlite and 11 for coherent kimberlite. The representative analyses are shown in Table 1, and the remaining analyses are presented in the Supplement Tables and Table 1. The index C.I. = $(SiO_2 + Al_2O_3 + Na_2O)/(MgO + K_2O)$ ranges from 0.95 to 1.2, whereas the index ilm. I = $(FeO_{tot} + TiO_2)/(2K_2O + MgO)$ is found to range between 0.21 and 0.44.

The contamination indices (Supplement Tables, Table S1) show that all the samples are relatively fresh.

**Table 1.** Representative chemical compositions of kimberlites from the Obnazhennaya pipe and Velikan dyke (in wt. %).

| | 1 | 2 | 3 | 4 | 5 | 6 | 7 | 8 | 9 | 10 |
|---|---|---|---|---|---|---|---|---|---|---|
| | 7-234 | 7-237 | 7-387 | 7-384 | 7-386 | 7-388 | 7-191 | 7-192 | 7-196 | 3 (7) |
| $SiO_2$ | 33.25 | 31.16 | 32.01 | 24.56 | 20.67 | 32.34 | 22.11 | 19.87 | 16.54 | 22.76 |
| $TiO_2$ | 0.57 | 0.60 | 0.47 | 1.35 | 1.25 | 4.75 | 4.20 | 3.75 | 3.94 | 3.28 |
| $Al_2O3$ | 2.87 | 1.82 | 2.69 | 2.65 | 2.92 | 3.72 | 3.25 | 2.89 | 3.10 | 4.39 |
| $Fe_2O_3$ | 5.95 | 7.06 | 4.00 | 5.25 | 6.58 | 6.25 | 5.53 | 4.65 | 7.12 | 7.64 |
| $FeO$ | 2.40 | 1.26 | 3.60 | 3.43 | 1.43 | 4.1 | 5.73 | 5.85 | 3.95 | 3.74 |
| $MnO$ | 0.13 | 0.14 | 0.15 | 0.26 | 0.20 | 0.12 | 0.20 | 0.19 | 0.16 | 0.19 |
| $MgO$ | 32.67 | 32.22 | 30.78 | 25.25 | 20.43 | 28.71 | 22.14 | 19.09 | 13.30 | 22.23 |
| $CaO$ | 6.98 | 8.45 | 8.92 | 16.52 | 20.42 | 4.56 | 16.32 | 20.84 | 23.40 | 15.90 |
| $Na_2O$ | 0.11 | 0.08 | 0.15 | 0.15 | 0.19 | 0.15 | 0.13 | 0.09 | 0.23 | 0.28 |
| $K_2O$ | 0.69 | 0.73 | 1.00 | 1.35 | 1.14 | 2.28 | 1.12 | 0.48 | 0.16 | 0.86 |
| $P_2O_5$ | 0.49 | 0.79 | 0.64 | 0.83 | 1.12 | 0.17 | 0.92 | 1.65 | 1.10 | 0.89 |
| $H_2O$ | 7.84 | 8.23 | 8.46 | 4.61 | 5.13 | 8.62 | 6.59 | 6.30 | 5.34 | 5.44 |
| $CO_2$ | 5.48 | 6.64 | 7.01 | 12.98 | 16.04 | 3.58 | 10.00 | 13.07 | 20.64 | 11.8 |
| Total | 99.43 | 99.18 | 99.98 | 99.19 | 97.52 | 99.35 | 98.24 | 98.72 | 99.33 | 99.4 |

1–6: Obnazhennaya pipe; 7–10: Velikan dyke. PK–pyroclastic kimberlite; CK–coherent kimberlite. 1–3: PK; 4–10: CK. 10: Middle composition (data are taken from [3]). $Fe_2O_3$, $H_2O$ and $CO_2$ were determined by chemical methods.

The pyroclastic kimberlite composes the majority of the rock outcrop of the Obnazhennaya pipe. The kimberlite samples are characterized by high $SiO_2$ (29.8–35.9 wt. %, mean 32.8 wt. %) and MgO (29.2–37.4 wt. %, mean 33.4 wt. %), and relatively low $FeO_{total}$, $TiO_2$, $K_2O$ and CaO + $CO_2$ contents, with average values of 7.6, 0.5, 0.7 and 12.9 wt. %, respectively (Table 1), and belong to the low-Fe petrochemical type [5]. Coherent kimberlite is characterized by high $CaCO_3$ (average 26.9 wt. %) content, a wide range of $TiO_2$ (from 0.3 to 4.75 wt. %; average 2.9 wt. %) and $K_2O$ (from 0.83 to 1.35 wt. %; average 1.1 wt. %), low $SiO_2$ (average 26.4 wt. %) and MgO (average 26.7 wt. %) and belongs to Mg-Fe and Fe-Ti petrochemical types. Two samples of coherent micaceous kimberlites (or glimmerites) (samples No. 7-388, 7-390) show high levels of $SiO_2$, $TiO_2$ and $K_2O$ (with an average of 34.1, 4.7 and 2.5 wt. %, respectively) and belong to the Fe-Ti-type kimberlites. The compositions of the pyroclastic and coherent kimberlites fall into separate fields on the correlation binary graphs, exhibiting minimal overlap (Figures 4 and 5). A strong negative correlation between $SiO_2$, CaO and $TiO_2$ (Figure 4B,D), and a positive correlation between carbonate components (CaO + $CO_2$) and $TiO_2$, $P_2O_5$ and MnO (Figure 5A–C) were observed in our samples and are common for kimberlites worldwide [1,3].

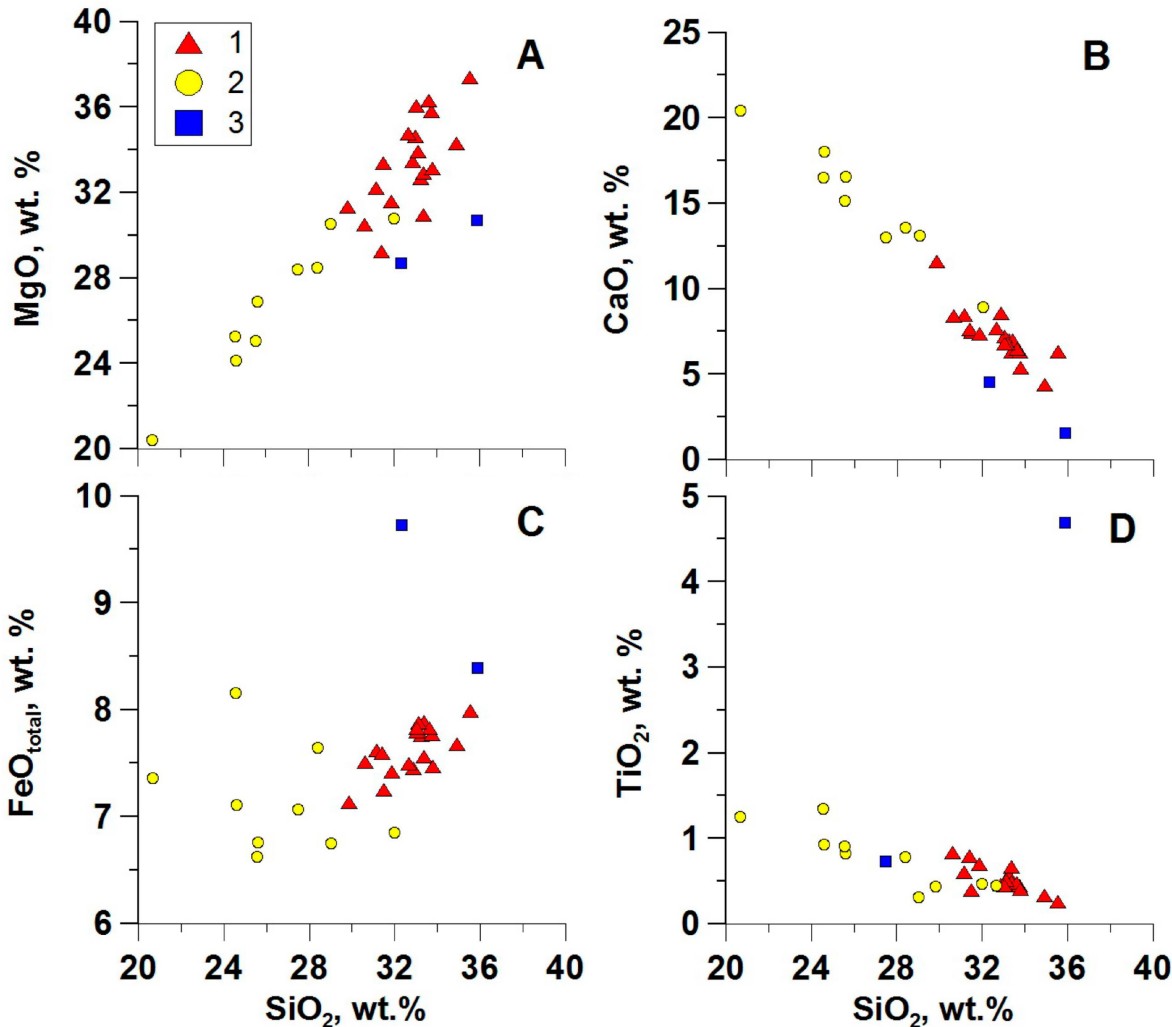

**Figure 4.** Correlation diagrams of $SiO_2$ vs. MgO, CaO, FeO and $TiO_2$ for Obnazhennaya kimberlite. 1—pyroclastic kimberlite; 2—coherent kimberlite; 3—inclusions of micaceous kimberlite. (**A**) $SiO_2$ vs. MgO; (**B**) $SiO_2$ vs. CaO; (**C**) $SiO_2$ vs. FeO (**D**) $SiO_2$ vs. $TiO_2$.

According to Table 1 and our previously published data [3], the chemical composition of the coherent kimberlite of the Velikan dyke is characterized by a high content of $FeO_{total}$ (10.0–10.7 wt. %) and $TiO_2$ (3.7–4.2 wt. %), which corresponds to Fe-Ti petrochemical type YaKP. Based on the $TiO_2$, $Al_2O_3$ and $FeO_{total}$ concentrations, the kimberlite in the Velikan dyke is similar to the sample 7-386 presented by coherent kimberlite of the Obnazhennaya pipe; however, it can be distinguished from the latter due to its elevated concentrations of MgO and $CaCO_3$.

The pyroclastic and coherent varieties of kimberlite from the Obnazhennaya pipe differ in both major and trace element compositions (Tables 1 and 2; Supplement Tables, Table S1). The pyroclastic kimberlite is characterized by a high Mg composition, as well as high concentrations of Ni, Co and Cr (Figure 6). The coherent samples exhibit elevated concentrations of incompatible elements (U, Th, Sr, Ba, Rb, Zr, Nb, REE and others) (Table 2) due to the increased content of the carbonate, alkaline and titanium components, when compared with the pyroclastic kimberlite.

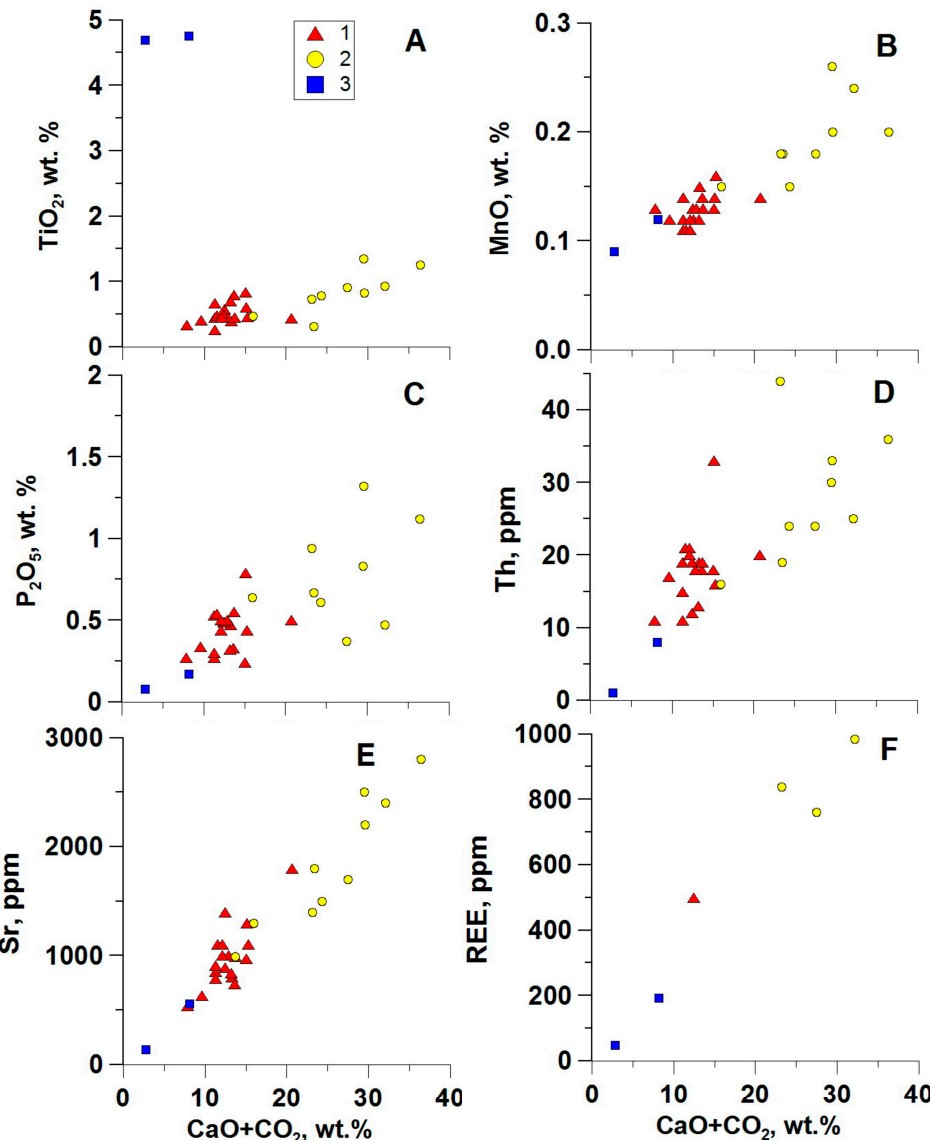

**Figure 5.** Correlation diagrams of carbonate component vs. major oxides and trace elements for Obnazhennaya kimberlite. The legend is from Figure 4. (**A**) CaO + CO$_2$ vs. TiO$_2$; (**B**) CaO + CO$_2$ vs. MnO; (**C**) CaO + CO$_2$ vs. P$_2$O$_5$; (**D**) CaO + CO$_2$ vs. Th; (**E**) CaO + CO$_2$ vs. Sr; (**F**) CaO + CO$_2$ vs. REE summa.

Based on the oxides and trace element correlation diagrams of the kimberlites from the Obnazhennaya pipe, we have divided them into three groups. The first group is distinguished by the highest degree of direct correlation and includes oxides like CaO, CO$_2$, MnO and P$_2$O$_5$, as well as incompatible elements (U, Th, Pb, Sn, Ag, Sc, Zr, Nb, Ta, Hf, Sr, Ba, REE, Y). The correlation coefficient of the carbonate component with these elements, as a rule, exceeds the value of 0.9. A high correlation level indicates that the carbonate component is the main concentrator of the incompatible elements (IE) (Figure 5D–F). The second group includes the major oxides (SiO$_2$, MgO) and trace elements (Ni, Co, Cr and B). The correlation coefficient between the elements and oxides is 0.6–0.8 for Ni, Co and B and 0.41 for Cr. The third group is presented by kimberlites that display a significant level of correlation between FeO$_{total}$, TiO$_2$, K$_2$O, Al$_2$O$_3$, Cs, Rb, Cu and V. In this group, rare elements are concentrated in Fe-Ti oxide minerals and phlogopite. The positive correlation between SiO$_2$ and FeO$_{total}$ for the pyroclastic kimberlite (Figure 3C) reflects the saturation level of the kimberlite melt with the olivine macrocrysts. This dependence does not exist for hypabyssal kimberlites. Finally, it can be concluded that a higher correlation of the coherent

elements (Ni, Co, etc.) with MgO is characteristic for pyroclastic kimberlite (Figure 6) and IE with $CaCO_3$ for the hypabyssal varieties of kimberlites (Figure 5).

**Table 2.** ICP-MS analysis of trace element compositions of kimberlites from the Obnazhennaya pipe and Velikan dyke (in ppm).

|  | 1 | 2 | 3 | 4 | 5 | 6 | 7 |
|---|---|---|---|---|---|---|---|
| **No.** | **7-234** | **7-242** | **7-384** | **7-386** | **7-388** | **7-390** | **7-191** |
| Sc | 11.6 | 12.9 | n.d. | 17 | n.d. | 4.93 | 22 |
| V | 75 | 74 | 111 | 143 | 183 | 181 | 397 |
| Cr | 1644 | 1073 | 858 | 400 | 1057 | 1975 | 548 |
| Co | 82 | 68 | 69.4 | 46 | 110 | 60 | 82 |
| Ni | 1371 | 1027 | 833 | 407 | 1415 | 592 | 416 |
| Cu | 23 | 34 | 72.7 | 53 | 219 | 84 | 124 |
| Zn | 51 | 48 | 76.8 | 55 | 110 | 46 | 96 |
| Rb | 41 | 47 | 69 | 52 | 110 | 222 | 91 |
| Sr | 845 | 1395 | 1903 | 2094 | 435 | 109 | 1313 |
| Y | 11.5 | 22 | 22.6 | 24 | 5.9 | 1.39 | 29 |
| Zr | 94 | 138 | 148 | 183 | 55.6 | 22 | 297 |
| Nb | 151 | 273 | 294 | 324 | 139 | 56 | 1092 |
| Cs | 0.63 | 0.73 | 0.61 | 0.92 | 1.18 | 1.71 | 1.48 |
| Ba | 1455 | 1660 | 2405 | 2867 | 578 | 317 | 6123 |
| La | 139 | 239 | 207 | 279 | 54 | 13.7 | 301 |
| Ce | 231 | 377 | 367 | 455 | 89 | 23 | 524 |
| Pr | 23 | 41 | 33 | 46 | 8.4 | 2.2 | 51 |
| Nd | 79 | 136 | 112 | 151 | 28.2 | 7.5 | 176 |
| Sm | 10.6 | 18 | 14.9 | 20 | 3.8 | 0.97 | 24 |
| Eu | 2.67 | 4.92 | 3.9 | 5.2 | 1.03 | 0.26 | 5.4 |
| Gd | 7.5 | 11.8 | 12.1 | 14.2 | 3.36 | 0.79 | 17 |
| Tb | 0.54 | 0.7 | 1.26 | 1 | 0.34 | 0.06 | 1.33 |
| Dy | 3.47 | 6.1 | 5.26 | 7 | 1.32 | 0.35 | 7.8 |
| Ho | 0.53 | 0.95 | 0.8 | 1.08 | 0.2 | 0.06 | 1.13 |
| Er | 1.15 | 1.92 | 1.68 | 2.22 | 0.47 | 0.13 | 2.34 |
| Tm | 0.14 | 0.23 | 0.19 | 0.28 | 0.05 | 0.02 | 0.28 |
| Yb | 0.83 | 1.29 | 1.09 | 1.61 | 0.29 | 0.09 | 1.69 |
| Lu | 0.11 | 0.15 | 0.14 | 0.2 | 0.03 | 0.01 | 0.22 |
| Hf | 2.31 | 3.14 | 3.41 | 4.27 | 1.56 | 0.74 | 8.7 |
| Ta | 6.8 | 10.5 | 10.6 | 12.9 | 11.4 | 7.4 | 5.7 |
| Pb | 8.6 | 9.8 | 11.2 | 16 | 3.61 | 0.75 | 13.7 |
| Th | 22 | 37 | 29.9 | 42 | 7.01 | 1.63 | 70 |
| U | 4.1 | 6.2 | 5.88 | 7.9 | 1.45 | 0.31 | 8.7 |

1–6: samples from Obnazhennaya pipe; 7: samples from Velikan dyke.

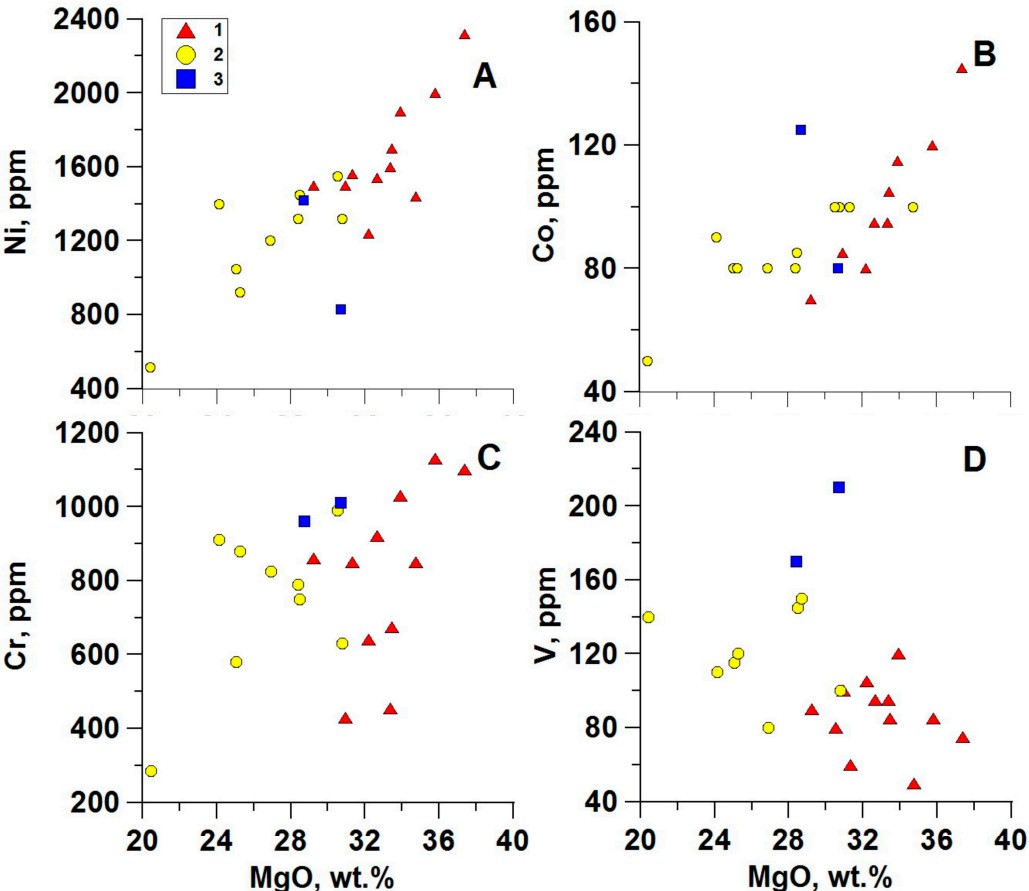

**Figure 6.** Correlation diagrams of MgO vs. Ni, Co, Cr and V for the Obnazhennaya kimberlite. The legend is from Figure 4. (**A**) MgO vs. Ni; (**B**)MgO vs. Co; (**C**) MgO vs. Cr; (**D**) MgO vs. V.

The trace element spider diagrams of the majority of the samples from the Obnazhennaya pipe (Figure 7) are similar to those observed in the kimberlitic rocks in YaKP [5]; however, there are two high Ti micaceous kimberlite inclusions (samples 7-388, 7-390) with positive anomalies in Cs, Rb, Nb, Ta and Ti, and V-shaped negative anomalies in Ba, U and Th. Both samples are strongly depleted in most incompatible elements, including REE. The rare element spectra for the Obnazhennaya hypabyssal kimberlites (samples 7-384, 7-386) are inconsistent with the typical field for diamondiferous kimberlite and record higher enrichment in incompatible elements relative to the pyroclastic kimberlite (sample 7-234).

The rare element composition of the Velikan dyke kimberlite (sample 7-191) differs from the Obnazhennaya pipe by a higher concentration of almost all the IE. This indicates that only coherent kimberlite from the Obnazhennaya pipe (sample 7-386), which has a comparable IE concentration to the kimberlite from the Velikan dyke, possesses the highest $CaCO_3$ content.

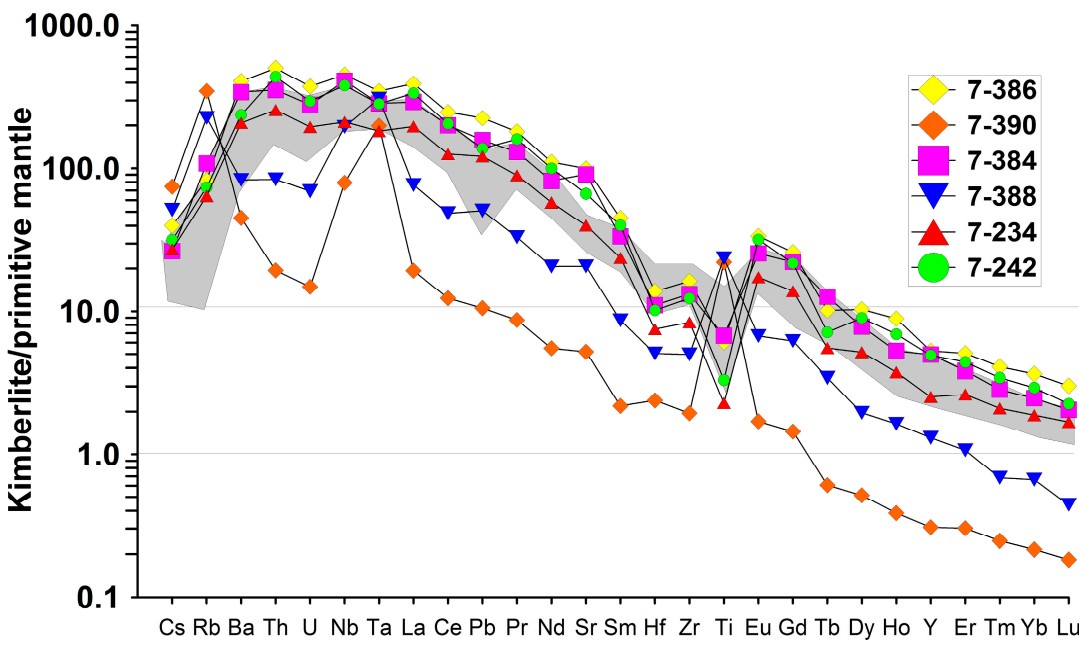

**Figure 7.** Trace element spider diagram for Obnazhennaya kimberlites. Gray-colored field corresponds to the kimberlites from diamond-bearing pipes [5].

*4.2. Sr-Nd-Hf Systematics*

Previously, we have studied the Sr and Nd isotope systematics of a representative collection of the YaKP kimberlites, including samples from Obnazhennaya [5]. In addition, we have analyzed Sr, Nd and Hf isotopic compositions of the kimberlites from different pipes of the Kuoika field, including the Obnazhennaya pipe and Velikan dyke, as well as diamondiferous pipes Udachnaya-East and International'naya from the Daldyn and Mirninsky fields (Table 3). The $(^{87}Sr/^{86}Sr)_i$–$\varepsilon Nd$ patterns for these kimberlites are typical of group I kimberlites and fall into the field of weakly depleted mantle close to the PREMA reservoir (Figure 8A). Deviation from PREMA toward more radiogenic Sr lies on a horizontal trend.

**Table 3.** Trace element abundances and Sr, Nd and Hf isotopic composition for kimberlites of Yakutian province. Rb, Sr, Sm, Nd, Lu, Hf are in ppm; age in Ma.

| Sample | Pipe | Field | Rb | Sr | $^{87}Rb/^{86}Sr$ | $^{87}Sr/^{86}Sr$ | 2σ | $(^{87}Sr/^{86}Sr)$ | Age |
|--------|------|-------|-----|------|--------|----------|----------|---------|-----|
| 00-289 | Inter | Mirniy | 19 | 689 | 0.0805 | 0.703856 | 0.000013 | 0.70344 | 360 |
| 03/33-1 | Udachnaya | Daldyn | 53 | 1510 | 0.1011 | 0.705995 | 0.000011 | 0.70548 | 360 |
| 03/91 | Udachnaya | Daldyn | 38 | 912 | 0.1199 | 0.705524 | 0.000012 | 0.70491 | 360 |
| 03-101 | Udachnaya | Daldyn | 59 | 1128 | 0.1525 | 0.707170 | 0.000012 | 0.70639 | 360 |
| 03-142 | Udachnaya | Daldyn | 70 | 1601 | 0.1263 | 0.707036 | 0.000012 | 0.70639 | 360 |
| 03-180 | Udachnaya | Daldyn | 91 | 1254 | 0.2096 | 0.706706 | 0.000014 | 0.70563 | 360 |
| 05-75 | Udachnaya | Daldyn | 31 | 1191 | 0.0749 | 0.705535 | 0.000013 | 0.70515 | 360 |
| 7-191 | Velikan | Kuoika | 85 | 1203 | 0.2037 | 0.704625 | 0.000015 | 0.70416 | 160 |
| 7-483 | Zenit | Kuoika | 15 | 926 | 0.0478 | 0.704116 | 0.000014 | 0.70401 | 160 |
| 7-487 | Jila 87/2 | Kuoika | 3 | 640 | 0.0130 | 0.705031 | 0.000013 | 0.70500 | 160 |
| 7-234 | Obnazhennaya | Kuoika | 39 | 828 | 0.1358 | 0.703924 | 0.000015 | 0.70362 | 160 |
| 7-237 | Obnazhennaya | Kuoika | 31 | 1114 | 0.08119 | 0.704361 | 0.000014 | 0.70418 | 160 |

**Table 3.** *Cont.*

| Sample | Pipe | Field | Rb | Sr | $^{87}$Rb/$^{86}$Sr | $^{87}$Sr/$^{86}$Sr | 2σ | ($^{87}$Sr/$^{86}$Sr) | Age |
|---|---|---|---|---|---|---|---|---|---|
| 7-242 | Obnazhennaya | Kuoika | 48 | 1512 | 0.0922 | 0.705836 | 0.000011 | 0.70563 | 160 |
| 7-280 | Obnazhennaya | Kuoika | 34 | 866 | 0.1140 | 0.706382 | 0.000015 | 0.70612 | 160 |
| 7-390 | Obnazhennaya | Kuoika | 216 | 106 | 5.9198 | 0.718436 | 0.000011 | 0.70497 | 160 |
| 7-390 | Obnazhennaya | Kuoika | 220 | 104 | 6.1251 | 0.718206 | 0.000012 | 0.70427 | 160 |
| 7-392 | Obnazhennaya | Kuoika | 30 | 920 | 0.0944 | 0.704303 | 0.000013 | 0.70409 | 160 |
| **Sample** | **Pipe** | **Sm** | **Nd** | $^{147}$**Sm/**$^{144}$**Nd** | $^{143}$**Nd/**$^{144}$**Nd** | **2σ** | **e$_{Nd}$(t)** | **2σ** | **Age** |
| 00-289 | Inter | 11.8 | 83.6 | 0.085221 | 0.512623 | 0.000015 | 4.84 | 0.29 | 360 |
| 03/33-1 | Udachnaya | 11.0 | 85.4 | 0.078185 | 0.512573 | 0.000009 | 4.18 | 0.18 | 360 |
| 03/91 | Udachnaya | 6.9 | 50.3 | 0.083196 | 0.512598 | 0.000012 | 4.44 | 0.23 | 360 |
| 03-101 | Udachnaya | 8.0 | 62.4 | 0.077734 | 0.512552 | 0.000011 | 3.80 | 0.21 | 360 |
| 03-142 | Udachnaya | 10.2 | 78.2 | 0.078820 | 0.512559 | 0.000011 | 3.89 | 0.21 | 360 |
| 03-180 | Udachnaya | 5.1 | 36.9 | 0.082929 | 0.512464 | 0.000013 | 1.85 | 0.25 | 360 |
| 7-191 | Velikan | 18.0 | 126.2 | 0.086474 | 0.512715 | 0.000011 | 3.76 | 0.21 | 160 |
| 7-483 | Zenit | 24.8 | 179.3 | 0.083792 | 0.512719 | 0.000011 | 3.89 | 0.21 | 160 |
| 7-487 | Jila 87/2 | 7.9 | 52.4 | 0.091687 | 0.512767 | 0.000015 | 4.67 | 0.29 | 160 |
| 7-234 | Obnazhennaya | 9.5 | 70.0 | 0.081715 | 0.512708 | 0.000015 | 3.72 | 0.29 | 160 |
| 7-237 | Obnazhennaya | 14.1 | 105.9 | 0.080360 | 0.512746 | 0.000013 | 4.48 | 0.25 | 160 |
| 7-242 | Obnazhennaya | 18.2 | 134.4 | 0.081806 | 0.512724 | 0.000012 | 4.02 | 0.23 | 160 |
| 7-280 | Obnazhennaya | 6.4 | 47.7 | 0.081011 | 0.512698 | 0.000012 | 3.54 | 0.23 | 160 |
| 7-390 | Obnazhennaya | 0.9 | 6.6 | 0.081050 | 0.512701 | 0.000014 | 3.59 | 0.27 | 160 |
| 7-390 | Obnazhennaya | 0.9 | 6.6 | 0.081001 | 0.512725 | 0.000014 | 4.06 | 0.27 | 160 |
| 7-392 | Obnazhennaya | 10.5 | 77.8 | 0.081684 | 0.512729 | 0.000012 | 4.13 | 0.23 | 160 |
| **Sample** | **Pipe** | **Lu** | **Hf** | $^{176}$**Lu/**$^{177}$**Hf** | $^{176}$**Hf/**$^{177}$**Hf** | **2σ** | **ε$_{Hf}$(t)** | **2σ** | **Age** |
| 00-289 | Inter | 0.0785 | 3.810 | 0.0029 | 0.282810 | 0.000015 | 8.57 | 0.52 | 360 |
| 03/33-1 | Udachnaya | 0.0600 | 2.859 | 0.0030 | 0.282700 | 0.000015 | 4.65 | 0.53 | 360 |
| 03/91 | Udachnaya | 0.0537 | 2.471 | 0.0031 | 0.282712 | 0.000013 | 5.08 | 0.45 | 360 |
| 03-101 | Udachnaya | 0.0529 | 3.158 | 0.0024 | 0.282691 | 0.000012 | 4.48 | 0.43 | 360 |
| 03-142 | Udachnaya | 0.0698 | 3.468 | 0.0029 | 0.282688 | 0.000018 | 4.26 | 0.63 | 360 |
| 03-180 | Udachnaya | 0.0462 | 2.203 | 0.0030 | 0.282616 | 0.000019 | 1.69 | 0.69 | 360 |
| 05-75 | Udachnaya | 0.0807 | 3.418 | 0.0034 | 0.282686 | 0.000010 | 4.08 | 0.37 | 360 |
| 7-191 | Velikan | 0.1677 | 5.214 | 0.0046 | 0.282898 | 0.000012 | 7.48 | 0.44 | 160 |
| 7-483 | Zenit | 0.1529 | 8.036 | 0.0027 | 0.282884 | 0.000009 | 7.20 | 0.33 | 160 |
| 7-487 | Jila 87/2 | 0.0666 | 4.403 | 0.0022 | 0.282881 | 0.000013 | 7.13 | 0.45 | 160 |
| 7-234 | Obnazhennaya | 0.0863 | 2.039 | 0.0060 | 0.282850 | 0.000014 | 5.64 | 0.48 | 160 |
| 7-237 | Obnazhennaya | 0.0985 | 2.756 | 0.0051 | 0.282895 | 0.000023 | 7.34 | 0.83 | 160 |
| 7-242 | Obnazhennaya | 0.1463 | 3.101 | 0.0067 | 0.282863 | 0.000014 | 6.03 | 0.51 | 160 |
| 7-242R | Obnazhennaya | 0.1463 | 3.092 | 0.0067 | 0.282851 | 0.000016 | 5.59 | 0.57 | 160 |
| 7-280 | Obnazhennaya | 0.0640 | 1.647 | 0.0055 | 0.282820 | 0.000032 | 4.63 | 1.13 | 160 |
| 7-390 | Obnazhennaya | 0.0104 | 0.661 | 0.0022 | 0.282889 | 0.000016 | 7.41 | 0.56 | 160 |
| 7-390 | Obnazhennaya | 0.0104 | 0.680 | 0.0022 | 0.282857 | 0.000012 | 6.29 | 0.44 | 160 |
| 7-392 | Obnazhennaya | 0.0844 | 2.369 | 0.0051 | 0.282889 | 0.000019 | 7.11 | 0.67 | 160 |

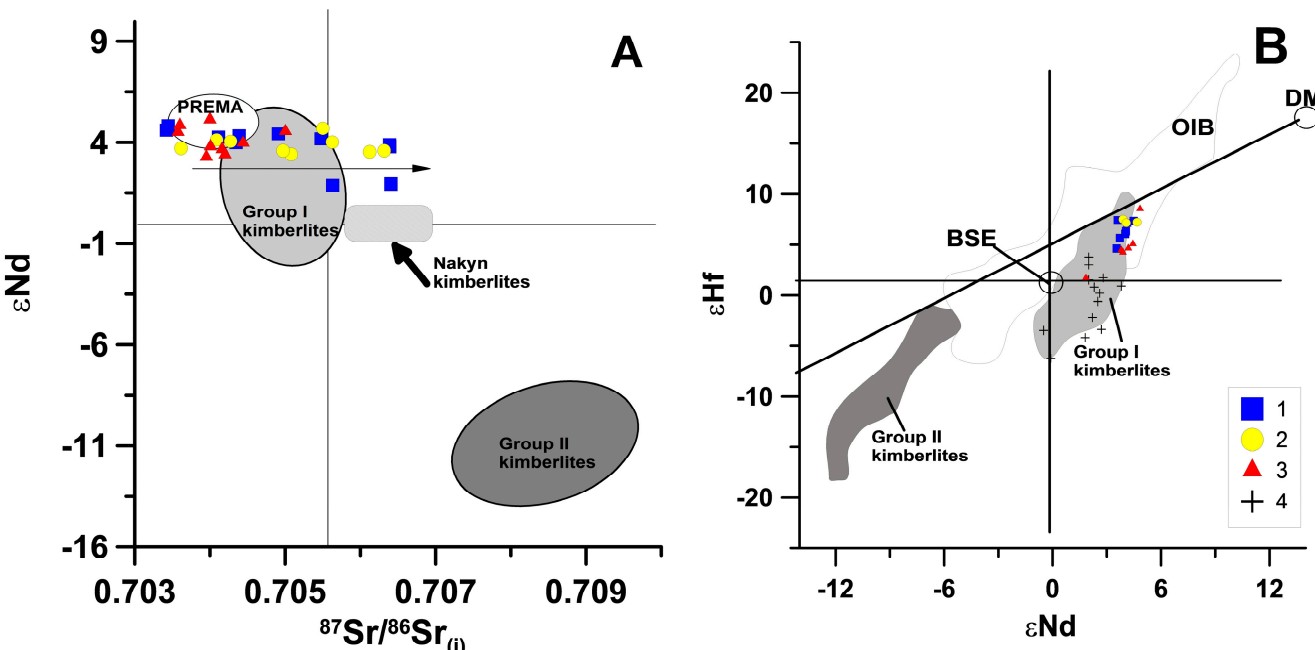

**Figure 8.** (**A**)—($^{87}$Sr/$^{86}$Sr)$_i$–εNd diagram for kimberlites from 1: diamond deposits, 2: Obnazhennaya pipe, 3: other pipes of the Kuoika field. Arrow marks the $^{87}$Sr/$^{86}$Sr evolution trend. Composition fields for kimberlites I and kimberlites II are after [22,51,52]. PREMA field is after [53]. (**B**)—εNd-εHf diagram for kimberlites from 1: diamond deposits, 2: Obnazhennaya pipe, 3: other pipes of Kuoika field, 4: South Africa [54]. Composition fields of group I and group II kimberlites are after [54].

Based on the Nd and Hf systematics, the kimberlites from the Kuoika and the diamondiferous kimberlites from the southern part of YaKP (Figure 8B) lie in the uppermost part of the field for South African kimberlites [54]. The YaKP data constitute a compact cluster located in the central OIB field and are predominantly concentrated within the mantle array.

### 4.3. Mineral Composition

We studied the composition of the minerals from the heavy mineral concentrate (HMC) of the kimberlites (in epoxy resin mounts), as well as the groundmass minerals (in anthills and thin sections). The HMC obtained from the eluvium of the pyroclastic kimberlite of the Obnazhennaya pipe contains olivine (85%–95%), garnet (5%–7%), spinel (3%–5%), clinopyroxene (2%–4%), ortopyroxene (1%–2%) and very rare grains of ilmenite. The HMC fine fraction (<0.2 mm) also contains sporadic euhedral grains of zircon and apatite. The low Fe pyroclastic kimberlite contains a low amount of magnetic minerals, even in the finest HMC fraction (–0.25). Macrocrysts are mostly sub-rounded or, less often, angular. The HMC from the Velikan dyke contains only microcryst olivine, phlogopite, apatite and opaque minerals (Ti–magnetite and perovskite).

Olivine from the Obnazhennaya pipe is presented by macrocrysts (>1 mm), groundmass microcrysts (<1 mm) and rock-forming minerals from the ultramafic mantle xenoliths. Representative analyses of olivine, variation ranges and the average values are given in Tables 4–6 and Supplementary Table S2. Olivine macrocrysts of the pyroclastic kimberlite are predominantly highly magnesian, and their composition generally corresponds to that of olivine from the peridotitic mantle xenoliths [9]. Coherent kimberlite comprises olivine macrocrysts that have compositions corresponding to those of xenolith olivine, as well as olivine with a higher FeO content.

**Table 4.** Variation ranges (numerator) and average contents (denominator) of major oxides in zoned Ol phenocrysts from the Obnazhennaya kimberlites (in wt. %).

| | **1** | **2** | **3** | **4** | **5** | **6** |
|---|---|---|---|---|---|---|
| | **237c (9)** | **7-237r (8)** | **7-387c (24)** | **7-387r (12)** | **386c (6)** | **7-386r (9)** |
| $SiO_2$ | 39.2–42.0 / 40.73 | 39.1–41.1 / 40.3 | 39.8–42.2 / 41.1 | 39.5–41.0 / 40.4 | 40.3–41.6 / 40.9 | 40.3–41.4 / 40.7 |
| $TiO_2$ | 0–0.01 / 0.01 | 0.01–0.04 / 0.02 | 0–0.02 / 0.01 | 0–0.05 / 0.02 | 0–0.04 / 0.02 | 0–0.04 / 0.03 |
| $Al_2O_3$ | 0–0.1 / 0.05 | 0–0.11 / 0.03 | 0–0.09 / 0.02 | 0–0.06 / 0.01 | 0–0.1 / 0.03 | 0–0.03 / 0.01 |
| $Cr_2O_3$ | 0–0.02 / 0.01 | 0.04–0.08 / 0.06 | 0–0.04 / 0.01 | 0.01–0.11 / 0.06 | 0–0.07 / 0.03 | 0.03–0.16 / 0.1 |
| $FeO$ | 7.04–8.0 / 7.51 | 11.8–12.9 / 12.3 | 6.5–10 / 7.5 | 11.9–13.6 / 12.4 | 7.7–10.3 / 9.0 | 11.2–12.0 / 11.5 |
| $MnO$ | 0.08–0.11 / 0.09 | 0.15–0.25 / 0.19 | 0.05–0.13 / 0.1 | 0.11–0.23 / 0.18 | 0.08–0.12 / 0.1 | 0.14–0.18 / 0.16 |
| $MgO$ | 45.4–51.2 / 49.05 | 45.3–47.5 / 46.4 | 48.6–51.7 / 50.4 | 45.6–47.2 / 46.5 | 47.9–51.0 / 49.5 | 46.5–48.4 / 47.7 |
| $CaO$ | 0–0.38 / 0.05 | 0.04–0.12 / 0.08 | 0–0.19 / 0.02 | 0.04–0.14 / 0.08 | 0–0.08 / 0.04 | 0.05–0.15 / 0.09 |
| $NiO$ | 0.35–0.42 / 0.38 | 0.08–0.27 / 0.18 | 0.34–0.42 / 0.38 | 0.09–0.32 / 0.21 | 0.32–0.42 / 0.36 | 0.24–0.39 / 0.32 |

1, 2: pyroclastic kimberlite; 3–6: coherent kimberlite; 1, 3, 5: core; 2, 4, 6: rim. The number of analyses is in brackets.

**Table 5.** The variation ranges and average contents (in the brackets) of major oxides in olivine from mantle xenoliths in Obnazhennaya kimberlites (according to [55]) (in wt. %). The number of analyses is in brackets.

| | **Lherzolite (21)** | **Olivine Websterites (13)** |
|---|---|---|
| $SiO_2$ | 39.8–42.1 (40.9) | 40.5–42.1 (41.0) |
| $TiO_2$ | 0–0.11 (0.02) | 0–0.06 (0.01) |
| $Al_2O_3$ | 0–0.04 (0.01) | 0–0.04 (0.01) |
| $Cr_2O_3$ | 0–0.04 (0.01) | 0–0.04 (0.01) |
| $FeO$ | 7.25–8.64 (8.0) | 6.95–8.41 (7.8) |
| $MnO$ | 0.06–0.11 (0.10) | 0.07–0.1 (0.09) |
| $MgO$ | 48.8–52.2 (50.4) | 49.1–51.6 (50.7) |
| $CaO$ | 0–0.03 (0.01) | 0–0.02 (0.01) |
| $NiO$ | 0.35–0.49 (0.4) | 0.37–0.44 (0.41) |
| Total | 98.8–101.7 (99.9) | 99.4–100.9 (100.2) |
| Mg# | 91.0–92.6 (91.8) | 91.4–92.9 (92.1) |

Groundmass olivine in the pyroclastic kimberlite varies from $10 \times 20$ to $400 \times 800$ μm in size; is oval, oval–angular or, rarely, euhedral microcrysts; and is partly serpentinized or carbonatized in the periphery or along microcracks. Many olivine grains show normal zoning (Table 4) with increased contents of FeO, $TiO_2$, $Cr_2O_3$, MnO and CaO, and a much lower content of NiO in the rims relative to the cores. Groundmass olivine in the coherent kimberlites is presented by euhedral or subhedral microcrysts (Figure 2C–E). The zoning olivine from the coherent kimberlites is similar to that from the pyroclastic kimberlites (Table 4; Supplementary Tables, Table S2), but some olivine cores are more ferrous compared

to olivine from the pyroclastic kimberlite (samples 7-386, 7-392). The NiO content in the rims of zoned microcrysts from the coherent kimberlites is higher when compared to the grains from the pyroclastic kimberlite (Figure 9).

**Table 6.** Chemical composition of Ol from Velikan dyke (sample of kimberlite—7-193) (in wt. %).

|  | SiO₂ | FeO | MnO | MgO | CaO | NiO | Total |
|---|---|---|---|---|---|---|---|
| 1 | 40.33 | 12.21 | 0.12 | 45.64 | 0.06 | 0.35 | 98.83 |
| 2 | 39.91 | 12.80 | 0.13 | 45.26 | 0.00 | 0.26 | 98.27 |
| 3 | 39.67 | 13.02 | 0.14 | 46.41 | 0.06 | 0.28 | 99.65 |
| 4 | 40.33 | 12.74 | 0.15 | 46.00 | 0.07 | 0.28 | 99.67 |
| 5 | 39.92 | 12.12 | 0.13 | 46.72 | 0.06 | 0.32 | 99.34 |
| 6 | 40.50 | 12.04 | 0.12 | 46.60 | 0.07 | 0.29 | 99.68 |
| 7 | 40.72 | 11.33 | 0.13 | 46.52 | 0.08 | 0.34 | 99.28 |
| 8 | 39.67 | 12.31 | 0.13 | 46.81 | 0.07 | 0.26 | 99.38 |
| 9 | 40.57 | 12.16 | 0.13 | 46.21 | 0.07 | 0.25 | 99.53 |
| 10 | 40.58 | 11.41 | 0.11 | 46.63 | 0.00 | 0.39 | 99.34 |
| 11 | 40.21 | 11.27 | 0.10 | 46.95 | 0.07 | 0.39 | 99.12 |
| Middle (11) | 40.22 | 12.13 | 0.13 | 46.34 | 0.05 | 0.31 | 99.28 |

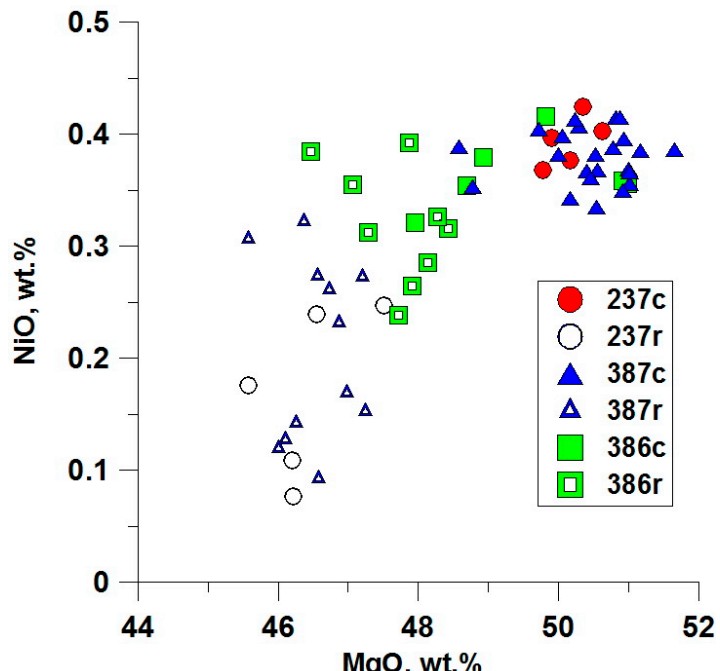

**Figure 9.** Correlation diagrams of MgO vs. NiO for zoned olivine macrocrysts and microcrysts from the Obnazhennaya pipe; c—core, r—rim. Olivine: 7-237, pyroclastic; 7-386 and 7-387, coherent kimberlite.

**Olivine** from the Velikan dyke is characterized (Table 6) by a relatively high Fe composition (FeO varies in a narrow range 11.3–13.0 wt. %). The NiO varies in the range of 0.25–0.39 wt. %.

**Garnet** from the HMC of the Obnazhennaya kimberlites has been characterized by 183 EMPA analyses (Supplementary Tables, Table S3). Based on the CaO–Cr₂O₃ plot, the macrocryst garnets (Supplementary Figure, Figure S3) lie in the relatively low Cr part of the lherzolite trend (up to 4 wt. % Cr₂O₃). Most garnets (96.7%) belong to one group,

conditionally called the pyroxenite–websterite. Garnets of dunite–harzburgite and wehrlite parageneses are absent. The compositions of Grt from the HMC and mantle xenoliths in the Obnazhennaya samples and reference data [9,10] are generally similar. Grt is absent in the Velikan dyke.

**Mg–ilmenite** is a very rare mineral in the HMC of the Obnazhennaya pipe and occurs as fine angular grains. Ilm (only 12 analyses) has an unusually low MgO (4.6–7.0 wt. %) and $Al_2O_3$ (0.11–0.17 wt. %; Supplementary Tables, Table S4) composition. Mg–ilmenite is absent in the Velikan dyke.

**Spinel** was studied in macrocrysts from HMC and the mantle xenoliths, including glimmerite. The pl macrocrysts in the Obnazhennaya kimberlite are rarer than Grt, but more abundant than ilm. The macrocrysts occur as grains from 0.1 to 1–2 mm in size, characterized by a xenomorphic angular shape, more rarely by a sub-rounded shape, and extremely rarely by an idiomorphic shape. Compositionally, Spl fits the isomorphic series of minerals: chromite ($FeCr_2O_4$)–Mg chromite ($MgCr_2O_4$)–spinel ($MgAl_2O_4$)–hercynite ($FeAl_2O_4$). The ulvöspinel and Mag minerals play subordinate roles: spl generally have low Ti and a weakly oxidized composition. Spl macrocrysts (Supplementary Tables, Table S5) exhibit broader ranges of $Cr_2O_3$ (19.2–52.3 wt. %) and $Al_2O_3$ (15.0–48.8 wt. %) when compared to the Spl from the pipes in the southern YaKP [8–10,56]. The Spl in the ultramafic mantle xenoliths is compositionally similar to the Spl macrocrysts from the kimberlites (Supplementary Figure, Figure S4).

**Clinopyroxenes** from the HMC occur as macrocrysts, usually angular, sometimes oval, and rarely of sub-idiomorphic shape. The grain size ranges up to 5–7 mm or greater. The composition of clinopyroxene macrocrysts (Supplementary Figure, Figure S5), according to the classification (Dawson, 1980), encompasses a wide range of species: sub-calcium diopside, diopside, Ti-Cr diopside, low-Cr diopside, high-Cr diopside and jadeite diopside. It has a broadly varying $Cr_2O_3$, $Al_2O_3$ and $Na_2O$ composition (0.18–3.2, 1.1–7.8 and 0.61–3.7 wt. %, respectively) and $Mg/(Mg + Fe) \times 100$ (Mg#) and $Ca/(Ca + Mg) \times 100$ (Ca#) ratios of 85.2–96.6 and 40.9–51.9, respectively (Table S6). The correlation diagrams $Mg/(Mg + Fe)$ and $Ca/(Ca + Mg)$ vs. $Cr_2O_3$, $Al_2O_3$ and $Na_2O$ (Supplementary Figure, Figure S5) demonstrate the similarity of clinopyroxene from the HMC with those from the mantle xenoliths. Clinopyroxene from micaceous kimberlite inclusions (or possibly glimmerites) differ from the other xenoliths by low Mg# and Ca# ratios of 87.2 and 47.9 on average, and by relatively low average contents of $Cr_2O_3$ and $Al_2O_3$ (0.81 and 2.82 wt. %).

According to the monomineral thermobarometer [57] and careful filtering the data, the clinopyroxene macrocrysts crystallized at pressures from 15 to 50 kbar (50–150 km) and temperatures from 700 to 1270 °C (the lower-temperature samples were discarded due to the thermometer characteristics) (Figure 10); however, the majority of the macrocrysts show a narrow PT range (15–35 kbar and 700–900 °C). None of the Cpx grains has PT parameters of crystallization within the diamond stability field. Generally, Cpx from the mantle xenoliths likewise crystallized at low pressures and temperatures, and none of them was within the diamond stability field. The thermal perturbation processes in the base of the lithosphere, as well as at a depth of 50–100 km might lead to such a large scattering of the PT data [58].

The majority of the Cpx from the Obnazhennaya pipe was formed at high heat flow (40–45 mW/m$^2$), and this is significantly above the values estimated for heat flow for lithospheric mantle beneath both the Udachnaya pipe and NE part of the Siberian Craton in the Middle Paleozoic [59,60]. It is difficult to estimate the lithospheric thickness beneath the Obnazhennaya pipe, and we can only assume that it was lower than in the middle part of the Siberian Craton [58]. Cpx is absent in the Velikan dyke.

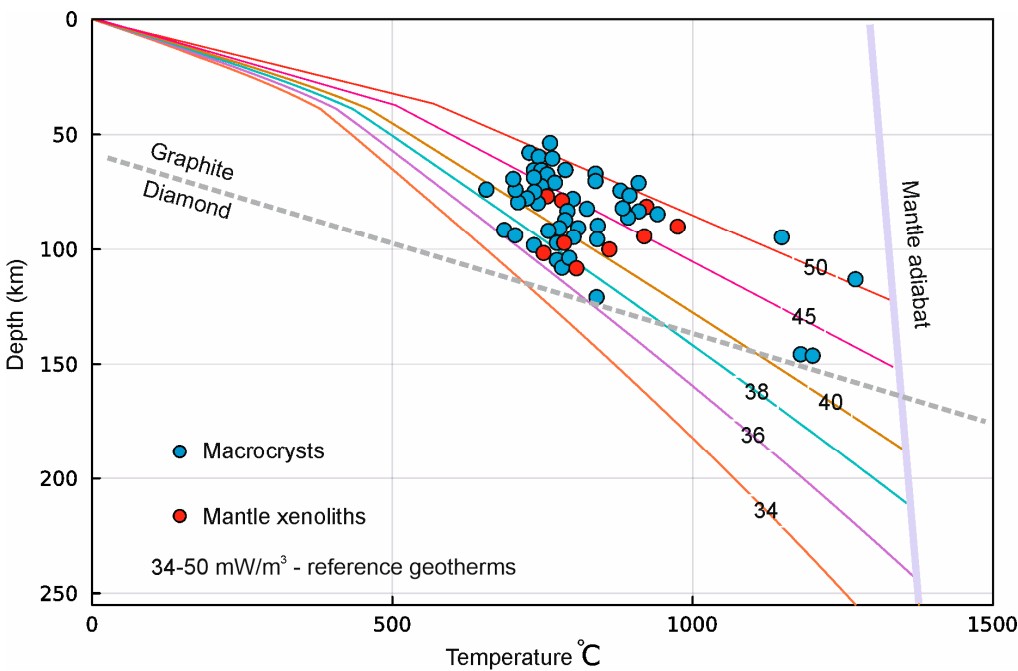

**Figure 10.** P-T diagram for clinopyroxenes from Obnazhennaya kimberlite (P-T estimates according to [57]). 1—macrocrysts, 2—mantle xenoliths. G-D—the line of graphite–diamond transformation.

**Phlogopite** occurs in pyroclastic kimberlite (sample 7-386) from the Obnazhennaya pipe either as isolated sporadic (1%–3%) macrocrysts with a lamellar structure ranging from 1-2 to 15 mm or as fine thin-lamellar oval crystals (from $50 \times 40$ to $100 \times 200$ μm) dispersed in the groundmass (Figure 2A–C). Coherent kimberlite (sample 7-386) contains phlogopite microcrysts in the groundmass (up to 15%–20%). These microcrysts are $50 \times 5$ to $250 \times 30$ μm in size and display oval, euhedral and subhedral shapes (Figure 2E). The preferred orientations of the phlogopites lead to a flow-alignment texture to the rock [45]. Phlogopite microcrysts from the coherent kimberlite (sample 7-386) exhibit significantly greater FeO and BaO contents when compared to pyroclastic kimberlite (samples 7-237, 7–7-387) (Supplementary Tables, Table S7). Macrocrysts of phlogopite are high Mg (Mg# ranges from 89.1–94.2) and low Ti (0.13–0.6 wt. %). The composition of phlogopite in the groundmass of the Velikan dyke kimberlites differs significantly from that in the Obnazhennaya pipe (Supplementary Tables, Tables S7 and S10; Supplementary Figure, Figure S6) and is characterized by a relatively high content of $TiO_2$ and BaO (ranges: 0.67–1.15 and 4.0–10.6 wt. % for the Velikan dyke; 0.1–4.4 and 0–1.0 for the Obnazhennaya pipe).

**Perovskite** occurs as isolated microcrysts ($20 \times 15$ to $70 \times 50$ μm in size) unevenly dispersed in the groundmass of coherent kimberlite from the Obnazhennaya pipe and Velikan dyke. Prv from the Obnazhennaya pipe differs from that in the Velikan dyke and has higher Ce and Nb and lower Nd contents (Supplementary Tables, Table S8). Prv microcrysts from the Velikan dyke show zoning with a decrease in Na, La, Ce and Nd content towards the rims, and an increase in Nb content.

**Carbonate** from the kimberlites of the Obnazhennaya pipe and Velikan dyke (Supplementary Tables, Table S9) is mainly calcite, less commonly dolomite and is present in the anhedral cryptocrystalline form. Calcite in the coherent kimberlites is found as euhedral tabular microphenocrysts that are commonly flow-aligned. The content of the Cal in the kimberlites of the Obnazhennaya pipe varies widely (ranging from 2.8 to 36.5%), being generally higher in the coherent kimberlite than in the pyroclastic kimberlite (15.1–36.5 vs. 7.8–20.7 %, respectively). Two samples of mica kimberlite with a massive texture (No. 7-388 and 7-390) have a very high $TiO_2$ content (up to 4.7 wt. %) and an extremely low calcite content (8.1 and 2.8%, respectively). Based on Figure 5, the correlation coefficients

show a very high level of direct correlation of the Cal content with MnO and $P_2O_5$ oxides and the majority of incompatible elements (U, Th, Pb, Ag, Sc, Zr, Nb, Ta, Hf, Sr, Ba, REE, Y). The carbonates have a MgO content ranging from 0.1 to 21.1 wt. % (Table S9). Quite often, calcite in the Obnazhennaya pipe and Velikan dyke contains elevated concentrations of SrO, reaching 0.89 and 0.67 wt. %, respectively.

A very rare earth mineral, ankelite [61], was found in the heavy mineral concentrate of the Velikan dyke. This mineral has the following composition: CaO (3.4), SrO (11.10), $La_2O_3$ (32.1), $Ce_2O_3$ (23.31) and $Nd_2O_3$ (0.68).

## 5. Discussion

Generally, the kimberlites from the northern fields of YaKP exhibit higher FeO, $TiO_2$ and $K_2O$ contents than the diamondiferous counterparts of the southern fields. Consequently, we classified most of the kimberlites from the northern fields into the Fe-Ti petrochemical type [5]. Thus, the Kuoika field is an exception, because the kimberlite bodies in this field are represented by both low-Fe (minority) and Fe-Ti (dominant) petrochemical types. These kimberlite compositions are similar to both northern and southern fields. The composition of the pyroclastic kimberlite in the Obnazhennaya pipe shows a similarity with the diamondiferous high-Mg kimberlites from the southern fields, while the kimberlites from the Velikan dyke are indicative of the Fe-Ti kimberlites found in the northern fields.

The main purpose of the article is to discuss the debatable issues about the variable chemical composition of the YaKP kimberlites. The problematic issues are as follows: different intrusion phases that differ in composition and the content of barophilic indicator minerals, including the rock-forming Ol; the primary composition of the kimberlite melt; the formation of the different petrochemical types of kimberlites; and the problem of the Ol crystallization from the kimberlite melt.

### 5.1. Primary Kimberlitic Melt (PKM)

Kimberlites are hybrid rocks that undergo various transformations during formation. These transformations significantly complicate the process of identifying the composition of PKM. An excellent review on this topic is provided in [18]. There are different points of view on the composition of PKM, which correspond to (1) the composition of an unaltered hypabyssal kimberlite [62–66]; (2) the composition of an aphanitic kimberlite that practically do not contain olivine macrocrysts [65–67]; (3) composition enriched in a chloride–carbonate component [67–69]; (4) an essentially carbonatite component [13]; (5) and the alkaline–basaltoid composition [70]. The fifth idea is based on the composition of the parent melt for a megacryst mineral association with a low Cr and high Ti content, the origin of which is genetically related to kimberlites [22,23,71]. Some researchers solve the problem of the composition of PKM by removing xenogenic Ol [64,72]. These studies suggest that PKM has a relatively high Mg composition containing 25–35 wt. % $SiO_2$ and MgO, and high CaO (12–20 wt. %) with variable concentrations of $H_2O$ and $CO_2$ [44,63–66].

Most attempts to decipher PKM fail to consider the substantial variations in the chemical composition of kimberlites across different provinces and even within different fields of the same province. These differences may be explained by the process of contamination by the lithospheric mantle rocks of different composition during the ascent of the kimberlite melt [17,21]. This idea is, in our opinion, correct but insufficient. Despite the relatively high Mg and low Ti composition of the lithospheric mantle rocks beneath the northern YaKP fields [39,73], the kimberlites of most northern fields, as we noted above, have a relatively high Fe and high Ti composition. Thus, the contamination process alone cannot explain the differences in the composition of the kimberlites and additional explanation is required. Our explanation is based on the presence of three types of inclusions of coherent kimberlite from the Obnazhennaya pipe and coherent kimberlite from the Velikan dyke. Since the coherent kimberlite from the Obnazhennaya pipe (all three of its varieties) contain olivine macrocrysts, its formation similar to the pyroclastic kimberlite was associated

with a process of assimilation of the lithospheric mantle material, although on a much smaller scale.

The lack of macrocrysts, including high-Mg olivine, in the coherent kimberlite of the Velikan dyke indicates that the corresponding PKM did not capture clastic material of the lithospheric mantle during its ascent and, therefore, retained its composition until it reached the surface. The composition of the low-Cr, high-Ti megacryst mineral associations is thought to be crystalized from the parental melt that has a composition similar to a high-Fe, high-Ti coherent kimberlite from the Velikan dyke (Figure 11). The amount of the clastic material assimilated by the kimberlite melts during the ascent might be also connected to the thickness of the lithosphere. In the central part of the Siberian Craton (Nakyn, Mirny, Alakit-Markha, Daldyn), the lithospheric thickness is much higher than in the margin (250–230 km vs. 180–200 km, Figures 10 and 11), and the amount harzburgitic rocks are also higher [39,58,60]. Thus, kimberlite assimilates harzburgites from a large volume of the lithospheric mantle, and the process of Opx dissolution is more effective, leading to a low Fe-Ti and "high-Mg" composition of the melts. Kimberlites with "high-Mg" characteristics are common in the central part of the Siberian Craton, where the Nakyn, Daldyn and Upper Muna fields are situated (Figure 11). The progressive thinning of the lithosphere from the central parts to the margins of the Craton is usually related to the plume activity by the time of Mesozoic kimberlite activity [58,60]; however, this issue should be further studied.

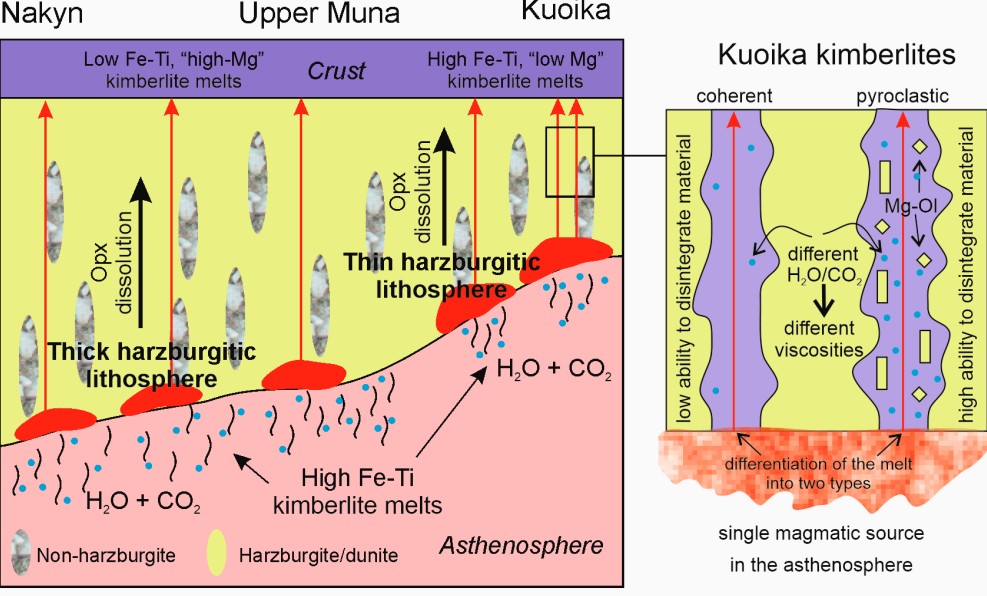

**Figure 11.** A model of kimberlite melts' formations. The differences in the composition of the kimberlites are associated with the following reasons: thickness of the lithospheric mantle, the amount of trapped xenogenic material with high MgO content (predominantly Opx), the concentrations of the $CaCO_3$ in the asthenospheric melts and the ability of these melts to assimilate the mantle material. The red arrows show kimberlite melts.

The spatial and temporal closeness and the identical isotope–geochemical characteristics of the kimberlites of the Obnazhennaya pipe and Velikan dyke let us assume the existence of a single magmatic source in the asthenosphere. In the previous sections of this article, the characterization of the composition of kimberlites from the YaKP reference kimberlite bodies is given. The maximum concentration levels of incompatible elements (IE) are found in coherent varieties of kimberlites, characterized by the highest content of the carbonate component (Table 1). The pyroclastic kimberlite of the Obnazhennaya pipe is characterized by a noticeably lower concentration of IE.

There is an opposite idea that the saturation of kimberlites with the carbonate component is the effect of a gradual increase in the amount of trapped CMLM, rather than to the process of melt differentiation. We cannot support this idea because the pyroclastic kimberlite of the Obnazhennaya pipe is characterized by the highest saturation of CMLM and the lowest carbonate content. The coherent kimberlite in the Velikan dyke with high carbonate content does not contain CMLM.

The data presented on the composition of Velikan dyke kimberlites also demonstrate the variations that reflect the process of melt differentiation during its ascent due to the process of gravitational separation of the Ol microcrysts; therefore, we estimate the composition of the primary asthenospheric melt as an average of the chemical composition of this kimberlite body, taking into account both our data and data from the literature [3] in wt. %: $SiO_2$—21.8, $TiO_2$—3.5, $Al_2O_3$—4.0, FeO—10.6, MnO—0.19, MgO—21.0, CaO—17.2, $Na_2O$—0.24, $K_2O$—0.78, $P_2O_5$—0.99, $CO_2$—12.6. The question of the $H_2O$ content in the primary melt remains debatable.

A comparison of the chemical, isotope–geochemical and mineral compositions of the kimberlite reference bodies suggests that the Velikan dyke's coherent kimberlite reflects the primary composition of the asthenospheric kimberlite melt. This primary composition represents the kimberlite melt that did not experience the process of contamination and partial assimilation of the CMLM.

### 5.2. Evolution of Kimberlites in Multi-Phase Pipes

Significant differences in the content of the detrital material were observed in the in pyroclastic kimberlite forming the peripheral and central parts of the Obnazhennaya pipe (up to 40%–50% and 10%–25%, respectively). Moreover, textural differences of the kimberlites from both parts point to the existence of individual, independent intrusion phases. Pyroclastic kimberlite from the peripheral zone contains clastic material with flora and fauna of the upper horizons and reflects the descending movement. The kimberlite of the central part contains xenoliths of the host rocks and a lot of xenoliths of the mantle rocks. Xenoliths are concentrated mainly in the columnar section of $3 \times 2$ m and display clear evidence of ascending movement. The inclusions of the coherent kimberlite within the pyroclastic kimberlite in the central part of the pipe indicate the existence of at least two independent stages of intrusion.

Pyroclastic kimberlite has low $TiO_2$ (0.3–0.8 wt. %, average 0.5 wt. %) and very low Ilm content, showing a clear similarity with the high-Mg petrochemical type (see "Analytical methods") of kimberlites [5], while the majority of autholiths of the coherent kimberlite refer to a Mg-Fe petrochemical type based on FeO (>9 wt. %) and $TiO_2$ (>1.5 wt. %) contents. Two autholiths of kimberlite saturated with Phl macrocrysts have very high $TiO_2$ (up to 4.7 wt. %) and $K_2O$ (up to 2.8 wt. %) contents corresponding to the Fe-Ti petrochemical type. The Obnazhennaya pipe, similar to the majority of the Yakutian pipes, is a multi-phase body [2,74,75]. The coherent kimberlites are presented only in the form of inclusions that unambiguously indicate their earlier stage of formation with respect to the host pyroclastic kimberlite. A higher content of MgO in the pyroclastic kimberlite (Table 1) shows the evolution of the composition toward the increasing MgO content. A similar evolutionary trend in the composition of the kimberlites was identified for other multiphase pipes situated in the YaKP (for example, Udachnaya-West, Udachnaya-East, Zarnitsa and Aikhal) [4,76].

### 5.3. How Was Pyroclastic Kimberlite Formed?

The origin of pyroclastic kimberlite is the key issue in understanding the formation mechanism of kimberlite rocks and the pipes. Previous studies [12,14,15,22,41,77] suggest that the root levels of the pipes are composed of coherent kimberlite, while the bulk of the kimberlite pipes is usually formed by pyroclastic kimberlites. According to this model, the source of pyroclastic kimberlite is a melt presented by coherent kimberlite, which experienced transformation due to alteration by hydrothermal fluids, phreatic explosions

and other processes occurring in the pipe or when the kimberlite melt approaches the pipe [22,24,25,31,78,79]. We suggest that this concept of the formation of pyroclastic kimberlites due to brecciation (or to a fluidization process) of coherent kimberlite does not apply to the central pyroclastic kimberlite in the Obnazhennaya pipe.

The generally accepted genetic conclusion on the origin of pyroclastic kimberlite concerning the hypabyssal (coherent) kimberlite transformation has not been supported by the analysis of the chemical and mineral composition of the corresponding kimberlite varieties in any of the studies on the classification issues [22–24,80]; however, there are a number of geological facts that disagree with this conclusion.

The study of kimberlite pipes of the YaKP with different levels of erosion (from 100–200 m to 1000 m and more) [34,81–83] and exploratory drilling data [5] showed that although the relative volume of coherent kimberlite increases with the depth of the pipes, pyroclastic kimberlites can be traced to the deepest root levels. The presence of pyroclastic kimberlite in the lower levels of the pipes is noted in model sections by other researchers [22,32,77,84–86]. Contact between the different kimberlite phases is usually sharp. We believe that the relationship between the coherent and some pyroclastic phases of kimberlites is more consistent with the idea of their individuality and independence of their formation from each other.

Differences in chemical and mineral composition between pyroclastic and coherent kimberlites from the Obnazhennaya pipe do not allow us to explain the origin of the former as a result of the transformation of the coherent kimberlite. The higher FeO content in the coherent kimberlites is consistent with the increased iron content in the Ol macrocryst and the inner parts of the Ol phenocrysts compared to those of the pyroclastic kimberlite (Table 4). The groundmass mineralogy also differs between the two types. Microcrysts of Phl from the coherent kimberlite of the Obnazhennaya pipe also differ from the Phl from the pyroclastic kimberlite in higher contents of FeO, $TiO_2$ and Ba (Supplementary Figures, Figure S9). Microcrysts of Prv and Ilm are commonly found in coherent kimberlites, whereas in the pyroclastic kimberlite, Prv is absent, and Ilm is very rare.

The most convincing evidence for the independent formation of coherent and pyroclastic varieties of kimberlite was obtained earlier [76] when studying the compositions of Ol macrocrysts from kimberlites of different intrusive phases in the Udachnaya-East pipe. Here, the initial coherent kimberlite phase is rich in high-Fe Ol macrocrysts (yellow-brown color). In contrast, the pyroclastic kimberlite has abundant high-Mg Ol macrocrysts (light green colour). High-Fe Ol macrocrysts were not found in the late stages of the pyroclastic kimberlite. It is clear that the composition of the rock-forming Ol is not changed (in part or in full) under the transformation of coherent kimberlite into pyroclastic.

We assume that the formation of the pyroclastic kimberlite with a massive texture was associated with the assimilation of the component that differs in composition from the phase corresponding to the coherent kimberlites (Figure 11). This phase is consistent in the FeO and $TiO_2$ contents and $H_2O$ and $CO_2$ ratio in the fluid component, which led to a higher ability of the melt to disintegrate the host rock (lithospheric mantle and crust) and in the manifestation of the fluidization process with breccia formation for pyroclastic kimberlite (Figure 11). Considering the leading role of fluidization processes in the formation of breccias [15,16,22,41,78,79,87], we note that, for some reason, the fluidization did not occur in melts corresponding to coherent kimberlites in the dykes and sills characterized by a high content of the carbonate component [88,89]. It is difficult to explain why fluidization occurs in pyroclastic kimberlite but does not occur in coherent kimberlite. Our suggestion is that the fluidization process was caused by the presence of $H_2O$ in the fluid. Unfortunately, we are not able to estimate the $CO_2/H_2O$ ratio in the fluid of primary melt for coherent and pyroclastic kimberlites. According to [90,91], $H_2O$ exists in the melt (however, it is not clear in what quantity). On the other hand, based on the fresh kimberlites (serpentinization is absent) from the Udachnaya-East pipe, other researchers [69] claim that the primary melt was anhydrous. An indirect argument for the role of $H_2O$ in the fluidization process can be found in the example of the Argyle lamproite pipe, which consists of diamond-bearing

tuffs [92]. It is assumed that the fluid phase of lamproites, in contrast to kimberlites, is dominated by $H_2O$ [93].

The beginning of the brecciation process should be attributed to the mantle depths. The saturation of the kimberlite melt with the Ol macrocrysts and mantle xenoliths is the beginning of the brecciation process [93]. It is noteworthy that mostly the mantle xenoliths in the Obnazhennaya pipe were concentrated in a small, column-shaped area, which indicates that they did not have a significant spread during the ascent and formation of pyroclastic kimberlite in the central part of the pipe. For the same reason, it is unacceptable to explain the origin of the pyroclastic kimberlite as a result of the return of the ejected kimberlite material into the pipe cavity [14].

As a rule, pyroclastic kimberlite tends to be more saturated with indicator minerals and mantle xenoliths than coherent kimberlite. This pattern is particularly evident when comparing kimberlites from the northern YaKP, especially the Kuoika field containing the Obnazhennaya pipe. In the Kuoika field (e.g., Monticellitovaya, Seraya, Velikan dyke, 87/2 dyke), there are separated kimberlite bodies formed only by coherent kimberlite, whose composition is enriched in FeO and $TiO_2$ [3]. The absence of mantle xenoliths and the extremely low content of indicator minerals (and sometimes a complete absence) in the HMC is a characteristic feature of the kimberlite bodies above. In this respect, some coherent kimberlite samples from the Obnazhennaya pipe (7-388 and 7-390) are close to the Velikan dyke in $FeO_{total}$ and $TiO_2$ contents: 9.3–10.8 wt. % and 4.7–4.75 wt. % (Table 1) against 10.5–12.2 wt. % and 3.3–3.7 wt. %, respectively [3].

An alternative explanation for the lack of macrocrysts in kimberlite from the Velikan dyke could be the process of the gravitational deposition of macrocrysts, leading to the formation of aphanitic varieties of kimberlite; however, aphanitic kimberlites usually have a relatively high Mg composition, which indicates that their formation was preceded by a process of the partial assimilation of the macrocrysts. Only the presence of varieties containing macrocrysts in these kimberlite bodies may confirm their genesis as a result of gravitational deposition. The high FeO composition of kimberlite from the Velikan dyke and the absence of macrocrysts in the kimberlites does not allow us to accept this explanation.

A high saturation with CMLM led to a higher value of Mg# for pyroclastic kimberlites from the southern diamondiferous fields of the YaKP, as well as higher diamond grades [4]. We assume that the formation of pyroclastic kimberlites, which have a massive texture of the groundmass, begins even when the kimberlite melt passes through the lithospheric mantle. Apparently, different phases of intrusion of kimberlites initially had different disintegrating abilities, predetermining a higher level of saturation with clastic material of the lithospheric mantle for pyroclastic kimberlite in comparison with coherent kimberlite. The origin of the kimberlite phases with different disintegrating abilities is associated with the differentiation of the asthenospheric melt that had occurred before it passed through the lithospheric mantle [71]. The existence of primary melts of different compositions during the formation of kimberlite pipes is also suggested by other researchers [64,94].

### 5.4. The Origin of Olivine in Kimberlites

Olivine, as the main rock-forming mineral, plays a crucial role in understanding the genesis of kimberlites. The conclusion that olivine macrocrysts are of xenogenic origin is incontrovertible for most researchers [13,26,27,68]. Recently, the debate has shifted to the question of the origin of olivine microcrysts. A number of researchers have expressed doubts about the possibility of isolating phenocryst olivine on the grounds that the olivine groundmass of kimberlites cores is, in their opinion, of xenogenic origin [26,27,68,80,95,96].

In pyroclastic varieties of kimberlite, the macrocrysts (excepting the rims) and mantle xenoliths do not differ in composition (Table 5), confirming the widely held view that they are predominantly xenogenic. On the other hand, in coherent varieties of kimberlites, the olivine fraction (macrocrysts and core portions of microcrysts) differs from the olivine of pyroclastic varieties with a higher FeO content (9.5–12.6 wt. %) (Supplement Tables, Table S2). Given the fact that the lithospheric mantle rocks under the Obnazhennaya pipe are

strongly dominated (approximately 95%) by a high-Mg olivine composition, we conclude that this part of olivine crystallized directly from the kimberlitic melt. The content of the relatively high-Fe mantle xenoliths in the Obnazhennaya pipe is extremely rare (less than 5% [9]; Tables 4 and 5); therefore, the pyroclastic kimberlite contains only high-Mg xenogenic Ol, whereas relatively high-Fe olivine macrocrysts have not been encountered.

The similarity of the compositions of olivine microcrysts and mantle xenoliths, together with the discovery of pyroxene and garnet inclusions in the cores of the olivine microcrysts, were the main arguments for concluding that olivine was of xenogenic origin; however, a detailed study of the compositions of zoned olivine and its crystalline inclusions by N.V. Sobolev et al. [97], using the example of unaltered kimberlite from the Udachnaya-East pipe, showed that olivine microcrysts differ from mantle olivine mainly in high Ti concentrations. It is noteworthy that clinopyroxene inclusions from zoned olivine kimberlites also differ from clinopyroxene from the mantle xenoliths. Based on the compositions [68,97], clinopyroxene inclusions are distinguished by higher values of the Ca/(Ca + Mg) ratio, indicating the low Ti nature of mineral crystallization, and bear no resemblance to xenogeneic clinopyroxene.

There are other indications of the potential for the crystallization of silicate minerals and, in particular, olivine and clinopyroxene, from the kimberlite melt. A low-Ti megacryst clinopyroxene intergrown with ilmenite was found in a kimberlite block of the Udachnaya-East pipe, unaltered by serpentinization, according to isotopic parameter ($^{87}Sr/^{86}Sr = 0.7029$ [28,98]), corresponding to the conclusion about its crystallization from the same source with kimberlites. Another argument is an ilmenite macrocryst form the HMC from the Udachnaya-East pipe, which contains an inclusion of ideally shaped faceted olivine crystal [99], which undoubtedly crystallized from a kimberlite melt. This suggests that not only olivine microcrysts, but also other silicate minerals have crystallized from the kimberlite melt [100].

The conclusion about the crystallization of some olivine in the kimberlite melt is confirmed by the study of the composition of olivine in the Velikan dyke, which contains only idiomorphic and subidiomorphic phenocrysts of high-Fe olivine (Table 6). It should be noted that high-Fe olivine of the euhedral form has also been established as the dominant mineral in the composition (Supplement Table S11; Supplementary Figure S7) of other kimberlite bodies of the northern YaKP fields, for example, in the Kuoika field (87/2 dyke, Montichelitovaya, Noyabrskaya, Haerdakh pipes) and Ary-Mastakh field (Beta, Rudniy Dvor dykes). All the kimberlites of these dykes and pipes are characterized by a high-Fe and high-Ti composition and belong to the Fe-Ti petrochemical type.

### 5.5. Mantle Sources of Kimberlites

On the ($^{86}Sr/^{87}Sr$)i-εNd plot (Figure 8A), the isotope characteristics of the kimberlites correspond to those of previously studied Type 1 kimberlites from different provinces around the world [15,22,51,52]. Most of the isotopic data points fall within the range of primitive and weakly depleted (near PREMA) mantle sources. The formation of the sub-horizontal trend on the ($^{87}Sr/^{86}Sr$)i-εNd plot has been explained [5] as a result of the intense secondary hydrothermal–metasomatic carbonatization process.

The Hf-Nd isotopic composition indicates a more depleted mantle source for the YaKP kimberlites compared to the South Africa kimberlites (Figure 8B) but shows strong similarity to the Neoproterozoic kimberlites from Baffin Island (Canada) [101].

The Kuoika kimberlites (including the Obnazhennaya pipe and the Velikan dyke) and the southern fields (the Udachnaya-East and Inter pipes) of YaKP have a similar distribution of Sr, Nd and Hf isotopic compositions (Figure 8A,B). This is especially interesting in view of their different ages of kimberlite emplacement events: in Devonia–n-Carboniferous (southern fields) and Jurassic (Kuoika field) time. The similarity of the Sr-Nd-Hf compositions for kimberlites over a large area in Yakutia (the diamond deposit is about 1000 km away from the Kuoika field) is evidence for isotopic homogeneity of the mantle source, which, in our opinion, should be of asthenospheric origin. The isotopic homogeneity of the

kimberlites of the YaKP is also accompanied by the geochemical homogeneity of the rocks, which is confirmed by the high level of similarity of the spidergrams for the distribution of incompatible elements for kimberlites from different fields of different ages of formation [5]. A clear deviation from this consistent pattern has been reported for the deposits of the Nakyn field, located about 350 km NE of the Mirny field: isotope–geochemical features of their composition indicate that their mantle source belongs to the intermediate type (between the sources characteristic of Groups I and II kimberlites) [51,102].

Another controversial question is whether the plumes are the source of kimberlites [103,104]. The duration of kimberlite volcanism (from 410 to 160 Ma) is confirmed by age determinations using the U-Pb method [37–40]. The Sr-Nd systematics and trace element distributions of samples of kimberlite of different ages and perovskite from the groundmass above the YaKP show identical characteristics [5], indicating the uniformity of the primary melts. The isotope–geochemical homogeneity of kimberlites of different ages in the extended area of the YaKP (and therefore the mantle source) can only be attributed to the asthenosphere rather than the plume, which must assimilate rocks of different characteristics during ascent. The lack of temporal correlation between the epoch of kimberlite volcanism and trap magmatism [37,105] on the Siberian platform is an additional argument for this conclusion. The conclusion that the asthenosphere is the primary mantle source of kimberlite rocks is fully consistent with the reported source characteristics [37,40,101,105].

## 6. Conclusions

1.  The contrasting compositions of two closely spaced kimberlite bodies, the Obnazhennaya pipe and the Velikan dyke, have been studied. The Obnazhennaya pipe is filled with volcanoclastic and pyroclastic types of high-Mg kimberlite; the latter contains rare fragments of coherent previous intrusion phases (autoliths, according to [25]). The pyroclastic kimberlite is highly saturated with olivine, pyroxene, garnet macrocrysts and mantle xenoliths. The Velikan dyke consists of high-Fe, high-Ti kimberlite that does not contain any mantle clastic material (neither Ol xenocrysts nor mantle xenoliths).

2.  In the multiphase Obnazhennaya pipe, three types of coherent kimberlite were found only in the pyroclastic kimberlite, indicating a later intrusion of the latter. Since a similar sequence of intrusion of different types of kimberlites was observed in most of the pipes we studied [4] (for example, in the pipes of Udachnaya-Western, Udachnaya-Eastern, Uybileynaya, Sytykanskaya, Aykhal, Komsomol'skaya and others), we believe that this regularity is general for kimberlite volcanism.

3.  The spatial proximity of the Obnazhennaya pipe and the Velikan dyke, the similarity of the Sr-Nd-Hf isotopic and trace element taxonomy (in terms of incoherent elements) for the corresponding kimberlites and, finally, the coincidence of their formation ages was the basis for concluding that they had a single magmatic asthenospheric source.

4.  The formation of compositionally contrasting kimberlites filling the Obnazhennaya pipe and the Velikan dyke was most likely due to the process of differentiation of the asthenospheric melt into two parts with different densities, viscosities and, consequently, disintegration capabilities. Their hypothetical composition is essentially carbonate and carbonate–silicate, probably characterized by different $H_2O$ contents. A melt of essentially carbonate composition, which had high integrability, formed a high-Mg petrochemical type of kimberlite; a carbonate–silicate melt formed Mg-Fe and Fe-Ti petrochemical types of kimberlite.

5.  The melt that formed the coherent kimberlite dyke of Velikan is, according to the authors, primary, as it does not contain xenogenic material from the lithospheric mantle and therefore has not been subjected to the process of its assimilation.

6.  A comparison of the composition of olivine from pyroclastic and coherent kimberlites, as well as from mantle xenoliths of the Obnazhennaya pipe showed that olivine from pyroclastic kimberlite is completely xenogenic; olivine from coherent types of kimberlite from the Obnazhennaya pipe is of both xenogenic and phenocryst origin;

and olivine from coherent kimberlites of the Velikan dyke is completely crystallized from the melt.

7. The geochemical homogeneity of the asthenospheric source under the YaKP, which persisted for a long time (410–160 Ma), is confirmed by the high level of similarity of the incompatible trace element patterns and the Sr-Nd-Hf isotopic systematics of the kimberlites for most of the kimberlites from different fields with different ages.

**Supplementary Materials:** The following supporting information can be downloaded at: https://www.mdpi.com/article/10.3390/min13111404/s1, Supplementary figures: Figure S1. A: A photo of outcrops of Obnazhennaya pipe. B: A photo of outcrops of Velikan dyke. Figure S2. A photo of thin section of sample 7-234. Obnazhennaya pipe (plane polarized light, PPL). Pyroclastic kimberlite with Cal-Srp groundmass, with relicts of fresh Ol. Figure S2-1. A photo of thin section of sample 7-243 (PPL). Obnazhennaya pipe. Pyroclastic kimberlite with inclusion of coherent kimberlite containing calcite microcrysts. Figure S2-2. A photo of thin section of sample 7-257 (PPL). Obnazhennaya pipe. Pyroclastic kimberlite with inclusion of coherent kimberlite containing calcite microcrysts. Figure S2-3. A photo of thin section of sample 7-325 (PPL). Obnazhennaya pipe. Coherent kimberlite with groundmass of essentially carbonate composition. Figure S3. CaO vs. $Cr_2O_3$ for garnet macrocrysts from Obnazhennaya pipe. Figure S4. Plots for spinel from Obnazhennaya pipe: A—of $Cr_2O_3$ vs. $Al_2O_3$, and B—of Mg/(Mg + Fe) × 100 vs. 1—kimberlite heavy fraction, 2—glimmerite, 3—peridotitic xenoliths. Figure S5. Plots: A—of Mg/(Mg + Fe) × 100 vs. $Cr_2O_3$; and B—Ca/(Ca + Mg) × 100 vs. $Cr_2O_3$ for clinopyroxenes from Obnazhennaya kimberlite. Clinopyroxenes are from 1—kimberlite heavy fraction, 2—mantle xenoliths, 3—glimmerite. Figure S6. Plots $TiO_2$-BaO and FeO-$Al_2O_3$ for groundmass phlogopite from Obnazhennaya kimberlite. Samples: 1—7-293 (pyroclastic kimberlite), 2—7-384 (coherent kimberlite), 3—7-386 (Phl coherent kimberlite). Figure S7(1–3). A photo of thin sections of coherent kimberlite samples from Velikan dyke. Figure S8. A photo of thin section of sample 90–20 from Rudniy dvor dyke (Ary-Mastakh field). Almost all olivine microcrysts are idiomorphic. Supplement tables: Table S1. Major oxide compositions of Obnazhennaya kimberlite. Table S2. Representative EMPA analyses of zoned microcryst Ol from Obnazhennaya kimberlites. Table S3. Compositions of macrocryst garnets from Obnazhennaya kimberlites. Table S4. Compositions of Mg–ilmenite from Obnazhennaya kimberlites. Table S5. Compositions of spinel macrocrysts from Obnazhennaya kimberlites. Table S6. Compositions of Cpx from Obnazhennaya kimberlites. Table S7. Representative EMPA analyses of Phl from Obnazhennaya and groundmass kimberlites Velikan dyke. Table S8. Representative analyses of perovskite from Obnazhennaya pipe and Velikan dyke, wt%. Table S9. Compositions of calcite and dolomite from Obnazhennaya pipe and Velikan dyke, wt%. Table S10. Compositions of minerals from Velikan dyke. Table S11. Composition of Ol from Kuoika and Ary-Mastakh fields.

**Author Contributions:** Conceptualization, S.K.; methodology, S.K.; formal analysis, S.K. and D.Y.; investigation, S.K. and I.A.; resources, S.K.; data curation, J.S., I.A. and O.B.; writing—original draft preparation, S.K., D.Y. and T.K.; writing—review and editing, S.K., T.K. and A.D.; visualization, D.Y. and A.D. All authors have read and agreed to the published version of the manuscript.

**Funding:** This research was funded by the Russian Science Foundation (Project 22-77-10073 "Reconstruction of the thermal state and composition of the lithospheric mantle beneath the kimberlite fields of the Siberian craton"). The work was performed using the scientific equipment of the Centre of Isotopic and Geochemical Research (Vinogradov Institute of Geochemistry SB RAS).

**Acknowledgments:** We thank the administration of the Amakinskaya and NIGP geological survey groups of the ALROSA Company for assistance with the subarctic field expeditions. The authors express their sincere gratitude to Sonja Aulbach for their attention, benevolence and many comments, which undoubtedly improved the text of the article and the reasonableness of the conclusions. The authors are grateful to Hilary Downes for her patience in reading the manuscript of the article and for correcting errors associated with the translation of the text of the article from Russian into English.

**Conflicts of Interest:** The authors declare no conflict of interest.

## Abbreviations

The following abbreviations are used in the paper: YaKP—Yakutian kimberlite province, CMLM—clastic material of the lithospheric mantle, HMC—heavy mineral concentrate, PKM—primary kimberlite melt, IE—incompatible elements, olivine—Ol, serpentine—Srp, chlorite—Chl, Mg–ilmenite—Ilm, garnet—Grt, spinel—Spl, clinopyroxene—Cpx, orthopyroxene—Opx, chromite—Chr, phlogopite—Phl, carbonate—Carb, apatite—Ap, magnetite—Mag, titanomagnetite—Timag, perovskite—Prv, zircon—Zrc.

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
