# Peer review of "Primary Composition of Kimberlite Melt"

_minerals, doi:10.3390/min13111404_

Round 1

Reviewer 1 Report

Comments and Suggestions for Authors

General comments:

-       Although I am not a native speaker, I am sorry to say that the first part of the manuscript is in poor English. Many sentences are unintelligible, some without verbs, sometimes with brackets that open but do not close. Incorrect syntax and poor grammar make some sections of the text almost unintelligible. The last few pages, however, are well written and the discussion is easy to follow.

-       Given the nature of kimberlites, and the high variability of crystals (both in size and type) rather than the random presence of xenocrysts and xenoliths - things also described by the authors - I think it is misleading to use whole rock analyses as representative of the kimberlite composition. The latter should be determined by analyses of what occupies the inter-grain spaces, or at least only of the groundmass. I know this is a point of view and as such is debatable, but it should at least be mentioned or briefly discussed in the paper.

-       Section 5.3 is one of the most interesting. I strongly recommend a figure that summarises your discussion, to help the reader follow your argument and visualise the model you propose.

-       Information about the studied samples is missing. How many samples were taken? How many thin sections were examined? … However, I appreciated the outcrops supplementary figures. If possible, you could also indicate there where the samples were taken from.

-       Tables and figures need more attention and consistency.

-       You presented a lot of useful and high-quality mineral chemistry data, why did you not discuss these data extensively in your paper? Please, at least expand the results section to include some references for comparison and possibly include some literature data in your mineral chemistry figures.

Specific comments:

47: do you mean “it contains kimberlites with high-Mg…”?

49: which contrasting compositions? Explain briefly

52: the sentence does not even include a verb

56: which “issues of kimberlite formation”? rephrase including the following sentences (line 57)

57: sentence without a verb

58: remains

58: which

64: which article? Yours?

65: this sentence is not understandable. What pipe are you talking about?

85: Yakutian

85: where does this rule come from? Cite the other occurrences.

90: do you mean with an area of 750 m2?

115: wide?

119: too much “or

121: rephrase

125: I would remove the word accessories

127: any description of these mantle xenoliths?

128: cylindric volume?

143: incomprehensible sentence

147: which rule?

148: glimmeritic

160: in figure3 any description in the caption is missing. Please add a short explanatory description as in figure2

165: I would prefer the term “sub-rounded in shape”. Anyway, how can a rounded (or oval) crystal be idiomorphic?

220: with whole rock analyses you will not get the composition of the kimberlite, but a sum of compositions of random xenoliths, xenocrysts, crystals, and kimberlite melt

270: no explanation on why some values are highlighted in green in the table

273: microelements??

286: figure3 is a thin section image not a graph

318: bibliography?

328: cannot see any error bars, are they smaller than the size of the dots in your graphs?

377: not comprehensible

396: not comprehensible

403: give some comparisons with literature data. You do this for garnets in line 402, as well as for Cpx and Ol, please give a literature reference here as well.

407: as stated above, please give some references for comparison. See for instance: Venier, M., Ziberna, L., Princivalle, F., Petrelli, M., Lughi, V., Logvinova, A., ... & Lenaz, D. (2022). Trace Elements in Chromian Spinels from Four Siberian Kimberlites. Minerals, 12(11), 1439.

444-458-464: same as above

479: what is the heavy kimberlite fraction? Do you mean the heavy minerals in it?

491: this is a crucial paragraph, please rephrase it here - as well as in the introduction -  to be clearer about the purpose of this work

521: what does it mean? It is not clear

572: something is missing in the sentence

593: this is one of the most interesting sections of your paper. I would suggest making a figure that summarises your model and helps the reader to follow your discussion

Comments on the Quality of English Language

Although I am not a native speaker, I am sorry to say that the first part of the manuscript is in poor English. Many sentences are unintelligible, some without verbs, sometimes with brackets that open but do not close. Incorrect syntax and poor grammar make some sections of the text almost unintelligible. The last few pages, however, are well written and the discussion is easy to follow. I consider a thorough revision of the text up to the discussion section to be necessary to make the work publishable. Also pay attention to the figure captions.

Author Response

We thank the reviewer very much for taking the time to review this manuscript, carefully reading and making detailed comments. Numerous comments made helped us to improve the manuscript. The responses and the corresponding revisions and changes are in the re-submitted files.

Questions for General Evaluation

The English language was significantly corrected and polished.

The large number of bulk kimberlite analyses allows to collect a large amount of statistics data, judge the melts composition and find a general trend (for example, Mg-rich kimberlites in the south of Yakutian province, Fe-rich in the north). Certainly, the reviewer is right, it is necessary to study the mineral composition of the interstitial mass for understanding the process. Unfortunately, in this case such delicate work was not carried out, we could only detected the mineral variety.   

The Figure 11 illustrating our model has been added. Thanks to the reviewer for this note.

We used 30 samples from the Obnazhennaya pipe and 10 samples from the Velikan dyke. The 20 thin sections were described. This information added to text.

Tables and figures have been corrected.

The section of mineral chemistry has been expanded and additional references inserted.

Reviewer 2 Report

Comments and Suggestions for Authors

The Kuoyka kimberlite field is located in an area of polychronous kimberlite magmatism (Middle Paleozoic, Late Jurassic) and shows many mysteries with no answers yet. Velikan dyke and Obnazhennaya pipe do not contain diamonds, but the same-age Dyanga pipe has a small number of diamonds. Late Jurassic kimberlite bodies also differ in mineralogy. The reasons for this are not yet precisely known. The authors of the paper chose two kimberlite bodies located close to each other but with different kimberlite composition and discuss possible reasons for this. The authors base their conclusions on a large amount of analytical material and this gives credibility to their conclusions. These conclusions can be disputed, but in the presence of a large number of questions about the origin of kimberlites of the Kuoyka field I believe that the hypothesis of the authors can be accepted. Therefore, the paper can be published as presented. Correction of the English language is necessary.

Comments on the Quality of English Language

Correction of the English language is necessary.

Author Response

We thank very much for benevolent review. Also many thanks to the reviewer for taking the time to review this manuscript, carefully reading and making detailed comments. We are also very grateful for the manuscript file with numerous edits, it really helped us improve the article. The correction version was resubmitted.

The English language was corrected and polished.

Reviewer 3 Report

Comments and Suggestions for Authors

This manuscript reports mineralogical characteristics and geochemical composition of two kimberlite bodies, the Obnazhennaya pipe and the Velikan dyke from the Kuoyka field, Yakutian kimberlite province. Overall, the manuscript is well written and the data are good. Thus, minor revision is required.

1)    Please supplement the description of geological characteristics of Yakutian kimberlite province.

2)    In the Figure 1, the longitude on the top is not clear. Please revise and number the two figures in Figure 1 as A and B.

3)    In caption of the Figure 2, please explain the abbreviations. Please make sure every single figure in the Figure 2 should be in the same size.

4)    In the Figure 3, please add the scale bar, and pay attention to the size of the picture.

5)    Please add units (wt. %) to the Table 1, and add units (ppm) to the Table 2.

6)    In the Table 2, where are the data of 8-10 samples from Velikan dyke? Please add the data.

7)    Please revise the symbol of celsius degree.

8)    Please change “5. Conclusions” into “6. Conclusions”.

9)    Please unify the formats of the Tables.

Comments on the Quality of English Language

Minor editing of English language required.

Author Response

We thank the reviewer very much for taking the time to review this manuscript, carefully reading, detailed comments and benevolent review. The comments made helped us to improve the manuscript. The responses and the corresponding revisions and changes are in the re-submitted files.

The English language was corrected and polished.

Reviewer 4 Report

Comments and Suggestions for Authors

Review of the manuscript ‘Contrast compositions of kimberlites - genetic conclusions 2 (Kuoyka field, Yakutia)’ by Kostrovitsky et al. The authors present mineral chemical, bulk-rock geochemical and Sr-Nd-Hf isotopic data on Jurassic kimberlites from Yakutia, Russia. The manuscript deals with an interesting research topic of broad scientific interest. A few flaws with the manuscript need to be fixed before it is considered for publication in Minerals. Some of the important comments are:

1. The title of the manuscript is not fit to describe the work that has been carried out. From the present title, it is hard to attract readers to the manuscript.

2. The Introduction section is the weakest in the manuscript. The research problem being addressed is not well described. This section should be general and should attract a wide range of readers. The introduction section and geological background are completely mixed, and I recommend re-writing this section.  

3. Though overall, the manuscript is well-written, still many sentences require reframing, and the abbreviation of minerals should be avoided in the text and table. The manuscript should be double-checked for language and grammar before it is considered for publication.

Since I am unaware of the regional geology, I have avoided commenting on that part. The rest of the comments are provided in a separate PDF file and can be easily fixed. I recommend the manuscript for publication after a moderate revision.

Comments on the Quality of English Language

Review of the manuscript ‘Contrast compositions of kimberlites - genetic conclusions 2 (Kuoyka field, Yakutia)’ by Kostrovitsky et al. The authors present mineral chemical, bulk-rock geochemical and Sr-Nd-Hf isotopic data on Jurassic kimberlites from Yakutia, Russia. The manuscript deals with an interesting research topic of broad scientific interest. A few flaws with the manuscript need to be fixed before it is considered for publication in Minerals. Some of the important comments are:

1. The title of the manuscript is not fit to describe the work that has been carried out. From the present title, it is hard to attract readers to the manuscript.

2. The Introduction section is the weakest in the manuscript. The research problem being addressed is not well described. This section should be general and should attract a wide range of readers. The introduction section and geological background are completely mixed, and I recommend re-writing this section.  

3. Though overall, the manuscript is well-written, still many sentences require reframing, and the abbreviation of minerals should be avoided in the text and table. The manuscript should be double-checked for language and grammar before it is considered for publication.

Since I am unaware of the regional geology, I have avoided commenting on that part. The rest of the comments are provided in a separate PDF file and can be easily fixed. I recommend the manuscript for publication after a moderate revision.

Author Response

We thank the reviewer very much for taking the time to review this manuscript, carefully reading and detailed comments. We are also very grateful for the manuscript file with edits, it really helped us improve the article. The correction version was resubmitted.

The English language was corrected and polished.

The title of manuscript was changed for more global description «Primary composition of kimberlite melt».

The section Introduction was changed and re-writing. 

The description of abbreviations added in the end of text, before the References section.
